# Model Selection for Off-policy Evaluation: New Algorithms and Experimental Protocol

**Pai Liu**
UIUC

**Lingfeng Zhao**
Columbia University

**Shivangi Agarwal**
IIIT Delhi

**Jinghan Liu**
USTC

**Audrey Huang**
UIUC

**Philip Amortila**
UIUC

**Nan Jiang**[*]
UIUC

## Abstract

Holdout validation and hyperparameter tuning from data is a long-standing problem in offline reinforcement learning (RL). A standard framework is to use off-policy evaluation (OPE) methods to evaluate and select between different policies, but OPE methods either incur exponential variance (e.g., importance sampling) or have hyperparameters of their own (e.g., FQE and model-based). We focus on model selection for OPE itself, which is even more under-investigated. Concretely, we select among candidate value functions ("model-free") or dynamics models ("model-based") to best assess the performance of a target policy. We develop: (1) new model-free and model-based selectors with theoretical guarantees, and (2) a new experimental protocol for empirically evaluating them. Compared to the model-free protocol in prior works, our new protocol allows for more stable generation and better control of candidate value functions in an optimization-free manner, and evaluation of model-free and model-based methods alike. We exemplify the protocol on Gym-Hopper, and find that our new model-free selector, LSTD-Tournament, demonstrates promising empirical performance.

## 1 Introduction

Offline reinforcement learning (RL) is a promising paradigm for applying RL to important application domains where perfect simulators are not available and we must learn from data [34, 23]. Despite the significant progress made in devising more performant *training* algorithms, how to perform holdout validation and model selection—an indispensable component of any practical machine learning pipeline—remains an open problem and has hindered the deployment of RL in real-life scenarios. Concretely, after multiple training algorithms (or instances of the same algorithm with different hyperparameter settings) have produced candidate policies, the *primary* task (which contrasts the *secondary* task which we focus on) is to select a good policy from these candidates, much like how we select a good classifier/regressor in supervised learning. To do so, we may estimate the performance (i.e., expected return) of each policy, and select the one with the highest estimated return.

Unfortunately, estimating the performance of a new *target* policy based on data collected from a different *behavior* policy is a highly challenging task, known as *off-policy evaluation* (OPE). Popular OPE algorithms can be roughly divided into two categories, each with their own critical weaknesses: the first is importance sampling [49, 22, 55], which has elegant unbiasedness guarantees but suffers variance that is *exponential* in the horizon, limiting applicability beyond short-horizon settings such as contextual bandits [36]. The second category includes algorithms such as Fitted-Q Evaluation (FQE) [14, 31, 47], marginalized importance sampling [38, 44, 57], and model-based approaches

---

[*]Correspondence to: nanjiang@illinois.edu.

39th Conference on Neural Information Processing Systems (NeurIPS 2025).

[61], which avoid the exponential variance; unfortunately, this comes at the cost of introducing their own hyperparameters (choice of neural architecture, learning rates, etc.). While prior works have reported the effectiveness of these methods [47], they also leave a chicken-and-egg problem: **if these algorithms tune the hyperparameters of training, who tunes *their* hyperparameters?**

In this work, we make progress on the latter problem, namely model selection for OPE algorithms themselves. We lay out and study a dichotomy of two settings: in the **model-based** setting, evaluation algorithms build dynamics models to evaluate a target policy. Given the uncertainty of hyperparameters in model learning, we assume that multiple candidate models are given, and the task is to select one that best evaluates the performance of the target policy. In the **model-free** setting, evaluation algorithms only output *value functions* which we select from; see Figure 1 (left) for a visualization of the pipeline. Notably, these selection algorithms should be *hyperparameter-free themselves*, to avoid further chicken-and-egg problems. Our contributions are:

1. **New model-free selector (Section 3):** We propose a new selection algorithm (or simply *selector*), LSTD-Tournament, for selecting between candidate value functions by approximately checking whether the function satisfies the Bellman equation. The key technical difficulty here is the infamous ***double sampling*** problem [7, 53, 10], which typically requires additional function approximation (hence hyperparameters) to bypass. Our derivation builds on BVFT [62, 64], which is the only existing selector that addresses double sampling in a theoretically rigorous manner without additional function approximation assumptions. Our new selector enjoys better statistical rates ($n^{-1/2}$ vs. $n^{-1/4}$) and empirically outperforms BVFT and other baselines.

2. **New model-based selectors (Section 4):** When comparing candidate models, popular losses in model-based RL (e.g., $\ell_2$ loss on next-state prediction) often exhibit biases under stochastic transitions [21]. Instead, we propose novel estimators with theoretical guarantees, including novel adaptation of previous model-free selectors that require additional assumptions, where the assumptions are automatically satisfied using additional information in the model set [6, 65].

3. **New experiment protocol (Section 5)**: To empirically evaluate the selection algorithms, prior works often use FQE to prepare candidate Q-functions [64, 46], which suffer from unstable training[2] and lack of control in the quality of the candidate functions. We propose a new experiment protocol, where the candidate value functions are induced from variations of the groundtruth environment; see Figure 1 (right) for an illustration. This bypasses the caveats of FQE and allows for the computation of Q-values in an *optimization-free* and controllable manner. Moreover, the protocol can also be used to evaluate and compare estimators for the model-based setting. Implementation-wise, we use ***lazy evaluation*** and Monte-Carlo roll-outs to generate the needed Q-values. Combined with parallelization and caching, we reduce the computational cost and make the evaluation of new algorithms easier.

4. **Preliminary experiments (Section 6)**: We instantiate the protocol in Gym Hopper [9] and demonstrate the various ways in which we can evaluate and understand different selectors.

## 2  Preliminaries and Model Selection Problem

**Markov Decision Process (MDP).**  An MDP is specified by $(\mathcal{S}, \mathcal{A}, P, R, \gamma, d_0)$, where $\mathcal{S}$ is the state space, $\mathcal{A}$ is the action space, $P : \mathcal{S} \times \mathcal{A} \to \Delta(\mathcal{S})$ is the transition dynamics, $R : \mathcal{S} \times \mathcal{A} \to [0, R_{\max}]$ is the reward function, $\gamma \in [0, 1)$ is the discount factor, and $d_0$ is the initial state distribution. A policy $\pi : \mathcal{S} \to \Delta(\mathcal{A})$ induces a distribution over random trajectories, generated as $s_0 \sim d_0$, $a_t \sim \pi(\cdot|s_t)$, $r_t = R(s_t, a_t)$, $s_{t+1} \sim P(\cdot|s_t, a_t)$. We use $\Pr_\pi[\cdot]$ and $\mathbb{E}_\pi[\cdot]$ to denote such a distribution and the expectation thereof. The performance of a policy is defined as $J(\pi) := \mathbb{E}_\pi[\sum_{t=0}^\infty \gamma^t r_t]$, which is in the range of $[0, V_{\max}]$ where $V_{\max} := R_{\max}/(1 - \gamma)$.

**Value Function and Bellman Operator.**   The Q-function $Q^\pi \in \mathbb{R}^{\mathcal{S} \times \mathcal{A}}$ is the fixed point of $\mathcal{T}^\pi : \mathbb{R}^{\mathcal{S} \times \mathcal{A}} \to \mathbb{R}^{\mathcal{S} \times \mathcal{A}}$, i.e., $Q^\pi = \mathcal{T}^\pi Q^\pi$, where for any $f \in \mathbb{R}^{\mathcal{S} \times \mathcal{A}}$, $(\mathcal{T}^\pi f)(s, a) := R(s, a) + \gamma \mathbb{E}_{s \sim P(\cdot|s,a)}[f(s', \pi)]$. We use the shorthand $f(s', \pi)$ for $\mathbb{E}_{a' \sim \pi(\cdot|s')}[f(s', a')]$.

**Off-policy Evaluation (OPE).**  OPE is about estimating the performance of a given *target* policy $\pi$ in the real environment denoted as $M^\star = (\mathcal{S}, \mathcal{A}, P^\star, R, \gamma, d_0)$, namely $J_{M^\star}(\pi)$, using an offline dataset $\mathcal{D}$ sampled from a behavior policy $\pi_b$. For simplicity, from now on we may drop the $M^\star$

---

[2]For example, our preliminary investigation has found that FQE often diverges with CQL-trained policies [27], which is echoed by Nie et al. [46] in personal communications.

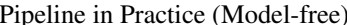

Pipeline in Practice (Model-free)          Experiment Protocol (Model-free)

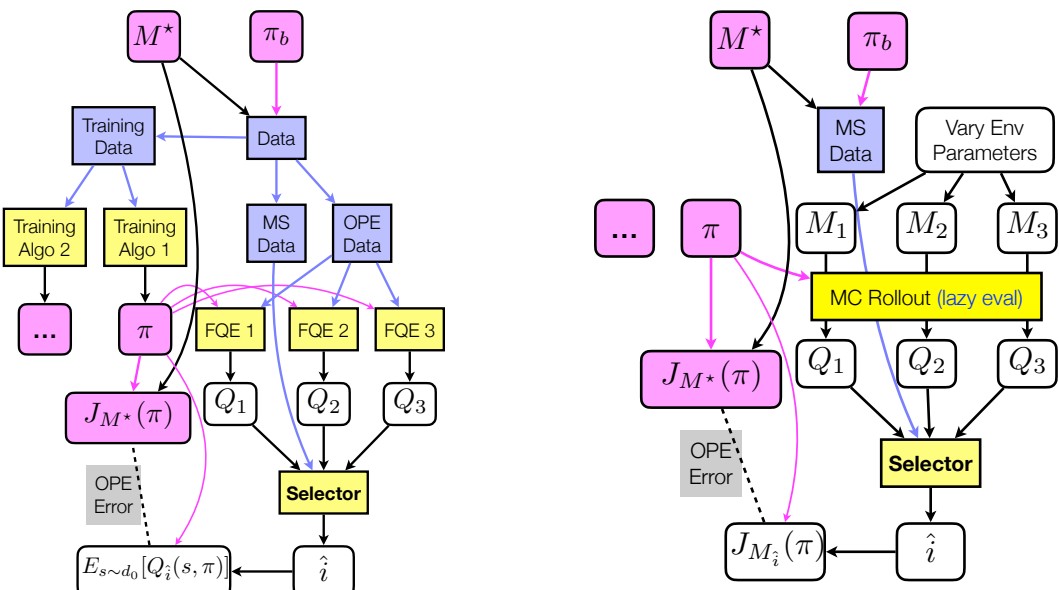

Figure 1: An illustration of the pipeline in practice that motivates our research (**left**) and our proposed experimental protocol (**right**), both for the model-free setting; the pipeline for the model-based setting is analogous and not visualized. **Left:** Training algorithms are run on training data to generate candidate policies, and choosing among them is the *primary model-selection (MS) task* (see Section 1) which is not the focus of our work. These policies (e.g., $\pi$) become the target policies in OPE (e.g., $\pi$), since accurate OPE can help solve the primary MS problem. Different FQE instances (e.g., with different hyperparameters, such as neural architectures) are used to approximate $Q^\pi$, producing $\{Q_i\}$. The **selector** takes MS data and $\{Q_i\}$ as input and choose one of them to estimate $J(\pi) \equiv J_{M^\star}(\pi)$. **Right:** Illustration of our protocol for an experiment *unit* (Section 5). The target policies $\pi$ can be learned from separate training data but can also be produced in other ways, such as training on inaccurate models $M_i$, which can be realistic for practical scenarios with inaccurate simulators. $\{M_i\}$ is prepared by varying environment parameters. Monte-Carlo rollouts are used to generate the Q-values for the data points in the MS data, which avoids potentially unstable optimization in the OPE pipeline; this is the source of stability and controllability compared to the prior protocol that mimics the practical pipeline. For further discussion on the limitation of our protocol and the trade-offs, see Section 7.

in the subscript when referring to properties of $M^\star$, e.g., $J(\pi) \equiv J_{M^\star}(\pi)$, $Q^\pi \equiv Q^\pi_{M^\star}$, etc. As a standard simplification, our theoretical derivation assumes that the dataset $\mathcal{D}$ consists of $n$ i.i.d. tuples $(s, a, r, s')$ generated as $(s, a) \sim \mu$, $r = R(s, a)$, $s' \sim P^\star(\cdot|s, a)$. We use $\mathbb{E}_\mu[\cdot]$ to denote the true expectation under the data distribution, and $\mathbb{E}_\mathcal{D}[\cdot]$ denotes the empirical approximation from $\mathcal{D}$.

**Model Selection.** We assume that there are multiple OPE *instances* (e.g. OPE algorithms with different hyperparameters) that estimate $J(\pi)$, and our goal is to choose among them based on the offline dataset $\mathcal{D}$; see Figure 1 for a visualization. Our setup is agnostic w.r.t. the details of the OPE instances, and views them only through the Q-functions or the dynamics models they produce.[3] Concretely, two settings are considered:

- **Model-free:** Each OPE instance outputs a Q-function. The validation task is to select $\hat{Q}$ from the candidate Q-functions $\mathcal{Q} := \{Q_i\}_{i \in [m]}$,[4] such that $\mathbb{E}_{s \sim d_0}[\hat{Q}(s, \pi)] \approx J(\pi)$. We are interested in the regime where at least one of the OPE instances produces a reasonable approximation of $Q^\pi$, i.e., $\exists i, Q_i \approx Q^\pi$, while the quality of other candidates can be arbitrarily poor. We make

---

[3]For alternative formulations such as selecting a function *class* for the OPE algorithm, see Appendix A.

[4]In practical scenarios, $\{Q_i\}_{i \in [m]}$ and $\{M_i\}_{i \in [m]}$ may be learned from data, and we assume $\mathcal{D}$ is a holdout dataset independent of the data used for producing $\{M_i\}_{i \in [m]}$ and $\{Q_i\}_{i \in [m]}$.

the simplification assumption $Q^\pi \in \mathcal{Q}$ for theoretical derivations, and handling misspecification ($Q^\pi \notin \mathcal{Q}$) is routine in recent RL theory and omitted for presentation purposes [3, 4]. We will also test our algorithms empirically in the misspecified case (Section 6.2).

- **Model-based:** Each OPE instance outputs an MDP $M_i$ and uses $J_{M_i}(\pi)$ as an estimate of $J(\pi)$. W.l.o.g. we assume $M_i$ only differs from $M^\star$ in the transition $P_i$, as handling different reward functions is straightforward. The task is to select $\hat{M}$ from $\mathcal{M} := \{M_i\}_{i \in [m]}$, such that $J_{\hat{M}}(\pi) \approx J(\pi)$. Similar to the model-free case, we assume $M^\star \in \mathcal{M}$ in the derivations.

The model-free algorithms have wider applicability, especially when we lack structural knowledge of the dynamics. A model-free algorithm can always be applied to the model-based setting: any $\{M_1, \ldots, M_m\}$ induces a Q-function class $\{Q^\pi_{M_1}, \ldots, Q^\pi_{M_m}\}$ which can be fed into a model-free algorithm (as we demonstrate in our protocol; see Section 5), and $M^\star \in \mathcal{M}$ implies $Q^\pi = Q^\pi_{M^\star} \in \mathcal{Q}$.

## 3 New Model-Free Selector

In this section we introduce our new model-free selector, LSTD-Tournament. To start, we review the difficulties in model-free selection and the idea behind BVFT [62, 64] which we build on.

### 3.1 Challenges of Model-free Selection and BVFT

To select $Q^\pi$ from $\mathcal{Q} = \{Q_1, \ldots, Q_m\}$, perhaps the most natural idea is to check how much each candidate function $Q_i$ violates the Bellman equation $Q^\pi = \mathcal{T}^\pi Q^\pi$, and choose the function that minimizes such a violation. This motivates the Bellman error (or residual) objective:

$$\mathbb{E}_\mu[(Q_i - \mathcal{T}^\pi Q_i)^2]. \tag{1}$$

Unfortunately, this loss cannot be estimated due to the infamous *double-sampling problem* [7, 53, 10], and the naïve estimation, which squares the TD error, is a biased estimation of the Bellman error (Eq.(1)) in stochastic environments:

$$\text{(TD-sq)} \qquad \mathbb{E}_\mu[(Q_i(s,a) - r - \gamma Q_i(s', \pi))^2]. \tag{2}$$

Common approaches to debiasing this objective involve additional "helper" classes (e.g., $\mathcal{G}$ in Section 4), making them not hyperparameter-free. See also Appendix A for a detailed discussion of how existing works tackle this difficulty under different formulations and often stronger assumptions.

**BVFT.** The idea behind BVFT [62] is to find an OPE algorithm for learning $Q^\pi$ from a function class $\mathcal{F}$, such that to achieve polynomial sample-complexity guarantees, it suffices if $\mathcal{F}$ satisfies 2 assumptions: (1) Realizability, that $Q^\pi \in \mathcal{F}$. (2) Some *structural* (as opposed to *expressivity*) assumption on $\mathcal{F}$, e.g., smoothness, linearity, etc. Standard learning results in RL typically require stronger expressivity assumption than realizability, such as the widely adopted *Bellman-completeness* assumption ($\mathcal{T}^\pi f \in \mathcal{F}, \forall f \in \mathcal{F}$). However, exceptions exist, and BVFT shows that *they can be converted into a pairwise-comparison subroutine* for selecting between two candidates $\{Q_i, Q_j\}$, and extension to multiple candidates can be done via a tournament procedure. Crucially, we can use $\{Q_i, Q_j\}$ to **automatically create an $\mathcal{F}$ needed by the algorithm** without additional side information or prior knowledge. We will demonstrate such a process in the next subsection.

In short, BVFT provides a general recipe for converting a special kind of "base" OPE methods into selectors with favorable guarantees. Intuitively, the "base" method/analysis will determine the properties of the resulting selector. The "base" of BVFT is $Q^\pi$-irrelevant abstractions [35, 20], where the structural assumption on $\mathcal{F}$ is being piecewise-constant. Our novel insight is that for learning $Q^\pi$, there exists another algorithm, LSTDQ [29], which satisfies the needed criteria and has superior properties compared to $Q^\pi$-irrelevant abstractions, thus can induce better selectors than BVFT.

### 3.2 LSTD-Tournament

We now provide a theoretical analysis of LSTDQ (which is simplified from the literature [42, 48]), and show how to transform it into a selector via the BVFT recipe. In LSTDQ, we learn $Q^\pi$ via linear function approximation, i.e., it is assumed that a feature map $\phi : \mathcal{S} \times \mathcal{A} \to \mathbb{R}^d$ is given, such that $Q^\pi(s,a) = \phi(s,a)^\top \theta^\star$, where $\theta^\star \in \mathbb{R}^d$ is the groundtruth linear coefficient. Equivalently, this asserts that the induced linear class, $\mathcal{F}_\phi := \{\phi(\cdot)^\top \theta : \theta \in \mathbb{R}^d\}$ satisfies realizability, $Q^\pi \in \mathcal{F}_\phi$.

LSTDQ provides a closed-form estimation of $\theta^\star$ by first estimating the following moment matrices:

$$\Sigma := \mathbb{E}_\mu[\phi(s,a)\phi(s,a)^\top], \quad \Sigma^{\mathrm{cr}} := \mathbb{E}_\mu[\phi(s,a)\phi(s',\pi)^\top], \tag{3}$$

$$A := \Sigma - \gamma\Sigma^{\mathrm{cr}}, \quad b := \mathbb{E}_\mu[\phi(s,a)r]. \tag{4}$$

As a simple algebraic fact, $A\theta^\star = b$. Therefore, when $A$ is invertible, we immediately have that $\theta^\star = A^{-1}b$. The LSTDQ algorithm thus simply estimates $A$ and $b$ from data, denoted as $\widehat{A}$ and $\widehat{b}$, respectively, and estimate $\theta^\star$ as $\widehat{A}^{-1}\widehat{b}$. Alternatively, for any candidate $\theta$, $\|A\theta - b\|_\infty$ can serve as a loss function that measures the violation of the equation $A\theta^\star = b$, which we can minimize over. Its finite-sample guarantee is given below. All proofs of the paper can be found in Appendix C.

**Theorem 1.** *Let $\Theta \subset \mathbb{R}^d$ be a set of parameters such that $\theta^\star \in \Theta$. Assume $\max_{s,a}\|\phi(s,a)\|_2 \leq B_\phi$ and $\max_{\theta\in\Theta}\|\theta\|_2 \leq 1$. Let $\widehat{\theta} := \arg\min_{\theta\in\Theta} \|\widehat{A}\theta - \widehat{b}\|_\infty$. Then, with probability at least $1 - \delta$,*

$$\|Q^\pi - \hat{\phi}^\top\theta\|_\infty \leq \frac{6\max\{R_{\max}, B_\phi\}^2}{\sigma_{\min}(A)}\sqrt{\frac{d\log(2d|\Theta|/\delta)}{n}}, \tag{5}$$

*where $\sigma_{\min}(\cdot)$ is the smallest singular value.*

Besides the realizability of $Q^\pi$, the guarantee also depends on the invertibility of $A$, which can be viewed as a coverage condition, since $A$ changes with the data distribution $\mu$ [2, 3, 23]. In fact, in the on-policy setting, $\sigma_{\min}(A)$ can be shown to be lower-bounded away from 0 [42]; see Appendix C.5.

**LSTD-Tournament.** We are now ready to describe our new selector. Recall that we first deal with the case of two candidate functions, $\{Q_i, Q_j\}$, where $Q^\pi \in \{Q_i, Q_j\}$. To apply the LSTDQ algorithm and guarantee, all we need is to create the feature map $\phi$ such that $Q^\pi$ is linearly realizable in $\phi$. In the spirit of BVFT, we design the feature map as

$$\phi_{i,j}(s,a) := [Q_i(s,a), Q_j(s,a)]^\top. \tag{6}$$

The subscript "$i,j$" makes it clear that the feature is created based on $Q_i$ and $Q_j$ as candidates, and we will use similar conventions for all quantities induced from $\phi_{i,j}$, e.g., $A_{i,j}, b_{i,j}$, etc. Obviously, $Q^\pi$ is linear in $\phi_{i,j}$ with $\theta^\star \in \{[1,0]^\top, [0,1]^\top\}$. Therefore, to choose between $Q_i$ and $Q_j$, we can calculate the LSTDQ loss of $[1,0]^\top$ and $[0,1]^\top$ under feature $\phi_{i,j}$ and choose the one with smaller loss. For $\theta = [1,0]^\top$, we have $A_{i,j}\theta - b_{i,j} =$

$$\mathbb{E}_\mu\left\{\begin{bmatrix}Q_i(s,a)\\Q_j(s,a)\end{bmatrix}([Q_i(s,a), Q_j(s,a)] - \gamma[Q_i(s',\pi), Q_j(s',\pi)])\right\}\begin{bmatrix}1\\0\end{bmatrix} \tag{7}$$

$$- \mathbb{E}_\mu[[Q_i(s,a), Q_j(s,a)]\cdot r]\begin{bmatrix}1\\0\end{bmatrix} = \mathbb{E}_\mu\left[\begin{bmatrix}Q_i(s,a)\\Q_j(s,a)\end{bmatrix}(Q_i(s,a) - r - \gamma Q_i(s',\pi))\right]. \tag{8}$$

Taking the infinity-norm of the loss vector, we have

$$\left\|A_{i,j}\begin{bmatrix}1\\0\end{bmatrix} - b_{i,j}\right\|_\infty = \max_{k\in\{i,j\}} |\mathbb{E}_\mu[Q_k(s,a)(Q_i(s,a) - r - \gamma Q_i(s',\pi))]|. \tag{9}$$

The loss for $\theta = [0,1]^\top$ is similar, where $Q_i$ is replaced by $Q_j$. Following BVFT, we can generalize the procedure to $m$ candidate functions $\{Q_1, \ldots, Q_m\}$ by pairwise comparison and recording the worst-case loss, this leads to our final loss function:

$$\mathcal{L}(Q_i; \{Q_j\}_{j\in[m]}, \pi) := \max_{k\in[m]} |\mathbb{E}_\mu[Q_k(s,a)(Q_i(s,a) - r - \gamma Q_i(s',\pi))]|. \tag{10}$$

The actual algorithm replaces $\mathbb{E}_\mu$ with the empirical estimation from data, and chooses the $Q_i$ that minimizes the loss. Building on Theorem 1, we have the following guarantee:

**Theorem 2.** *Given $Q^\pi := Q_{i^\star} \in \{Q_i\}_{i\in[m]}$, the $Q_{\widehat{i}}$ that minimizes the empirical estimation of $\mathcal{L}(Q_i; \{Q_j\}_{j\in[m]}, \pi)$ (Eq.(10)) satisfies that w.p. $\geq 1 - \delta$,*

$$|J(\pi) - \mathbb{E}_{s\sim d_0}[Q_{\widehat{i}}(s,\pi)]| \leq \max_{i\in[m]\setminus\{i^\star\}} \frac{24V_{\max}^3}{\sigma_{\min}(A_{i,i^\star})}\sqrt{\frac{\log(8m/\delta)}{n}}. \tag{11}$$

**Comparison to BVFT [62].** BVFT's guarantee has a slow $n^{-1/4}$ rate for OPE [64, 19], whereas our method enjoys the standard $n^{-1/2}$ rate. The difference is due to an adaptive discretization step in BVFT, which also makes its implementation somewhat complicated as the resolution needs to be heuristically chosen. By comparison, the implementation of LSTD-Tournament is simple and straightforward. Both methods inherit the coverage assumptions from their base algorithms and are not directly comparable, and a nuanced discussion on this issue can be found in Appendix C.5.

**Variants.** A key step in the derivation is to design the linearly realizable feature of Eq.(6), but the design is not unique as any non-degenerate linear transformation would also suffice. For example, we can use $\phi_{i,j} = [Q_i/c_i, (Q_j - Q_i)/c_{j,i}]$; the "diff-of-value" term $Q_j - Q_i$ has shown improved numerical properties in practice [27, 12], and $c_i, c_{j,i}$ can normalize the discriminators to unit variance for further numerical stability; this will also be the version we use in the main-text experiments. Empirical comparison across these variants can be found in Appendix E.2.

## 4 New Model-Based Selectors

We now consider the model-based setting, i.e., choosing a model from $\{M_i\}_{i\in[m]}$ such that $J_{\hat{M}}(\pi) \approx J(\pi)$. This can be practically relevant when we have structural knowledge of the system dynamics and can build reasonable simulators, but simulators of complex real-world systems will likely have many design choices and knobs that cannot be set from prior knowledge alone. As alluded to at the end of Section 2, model-free methods can always be applied to the model-based setting by letting $\mathcal{Q} := \{Q^\pi_{M_i}\}_{i\in[m]}$. The question is: *are there methods that leverage the additional side information in $\mathcal{M}$ (that is not in $\mathcal{Q}$) to outperform model-free methods?*

We study this question by developing algorithms with provable guarantees that can only be run with $\mathcal{M}$ but not $\mathcal{Q}$ (which means they must be using additional information), and include them in the empirical comparisons in Section 6. To our surprise, however, these algorithms are outperformed by LSTD-Tournament, which is simpler, computationally more efficient, and more widely applicable. Despite this, our model-based development produces novel theoretical results and can be of independent interest, and also provides additional baselines for LSTD-Tournament in empirical comparison. We briefly describe the studied methods below, with details deferred to Appendix B.

1. **Naïve Baseline.** A natural method is to use the model-prediction loss: any model $M$ is scored by $\mathbb{E}_{(s,a,s')\sim\mu, \tilde{s}\sim P(\cdot|s,a)}[\|s' - \tilde{s}\|]$, where $P$ is the dynamics of $M$, and $\|\cdot\|$ is some norm (e.g., $\ell_2$ norm) that measures the distance between states. Despite its wide use in the model-based RL literature, the loss has a crucial caveat that **it exhibits a bias towards more deterministic systems**, which is problematic when the groundtruth environment is stochastic. We will see empirical evidence of this in Section 6 (see "mb_naive" in Figures 3 and 4).

2. **Regression-based Selector (Appendix B.1).** Recall from Section 3 that a main difficulty in directly estimating the Bellman error is the lack of access to $\mathcal{T}^\pi$. Antos et al. [6] suggest that $\mathcal{T}^\pi f$ for any $f$ can be learned via regression $\mathcal{T}^\pi f \in \arg\min_{g\in\mathcal{G}} \mathbb{E}_\mu[(g(s,a) - r - \gamma f(s', \pi))^2]$, provided that $\mathcal{T}^\pi f \in \mathcal{G}$. Zitovsky et al. [65] apply this idea to the model selection problem and uses a user-provided $\mathcal{G}$, which requires additional hyperparameters as a model-free selector. However, in Appendix B.1 we show that the $\mathcal{G}$ class can be constructed *automatically* from $\mathcal{M}$, making it a strong baseline (due to its use of additional information in $\mathcal{M}$) for our LSTD-Tournament in the empirical comparison ("mb_Zitovsky et al." and "mb_Antos et al" in Figure 4).

3. **Sign-flip Selector (Appendix B.2).** We also develop a novel selector that takes the spirit of regression-based selector, but replace its squared loss with absolute value. It comes with new theoretical guarantee (Theorem 4), and is implemented in the experiments ("mb_sign_flip").

## 5 A Model-Based Experiment Protocol

Given the new selectors, we would like to evaluate and compare them empirically. However, as alluded to in the introduction, current experiment protocols have various caveats and make it difficult to evaluate the estimators in well-controlled settings. In this section, we describe a novel model-based experiment protocol, which can be used to evaluate both model-based and model-free selectors.

## 5.1 The Protocol

Our protocol consists of experiment textitunits defined by the following elements (see Figure 1R):

1. Groundtruth model $M^\star$.
2. Candidate model list $\mathcal{M} = \{M_i\}_{i \in [m]}$.
3. Behavior policy $\pi_b$ and offline sample size $n$.
4. Target policies $\Pi = \{\pi_1, \ldots, \pi_l\}$.

Given the specification of a unit, we will draw a dataset of size $n$ from $M^\star$ using behavior policy $\pi_b$. For each target policy $\pi \in \Pi$, we apply different selectors to choose a model $M \in \mathcal{M}$ to evaluate $\pi$. Model-free algorithms will access $M$ only through its Q-function, $Q_M^\pi$, effectively choosing from the set $\mathcal{Q} = \{Q_M^\pi : M \in \mathcal{M}\}$. Finally, the prediction error $|J_M(\pi) - J_{M^\star}(\pi)|$ is recorded and averaged over the target policies in $\Pi$. Moreover, we may gather results from multiple units that share the same $M^\star$ but differ in $\mathcal{M}$ and/or the behavior policy to investigate issues such as robustness to misspecification and data coverage, as we will demonstrate in the next section.

**Lazy Evaluation of Q-values via Monte Carlo.** While the pipeline is conceptually straightforward, practically accessing the Q-function $Q_M^\pi$ is nontrivial: we could run TD-style algorithms in $M$ to learn $Q_M^\pi$, but that invokes a separate RL algorithm that may require additional tuning and verification, and it can be difficult to control the quality of the learned function.

Our innovation here is to note that, for all the model-free algorithms we are interested in evaluating, **they all access $Q_M^\pi$ exclusively through the value of $Q_M^\pi(s,a)$ and $Q_M^\pi(s',\pi)$ for $(s,a,r,s')$ in the offline dataset** $\mathcal{D}$. That is, given $n$ data points in $\mathcal{D}$, we only need to know $2n$ scalar values about $Q_M^\pi$. Therefore, we propose to directly compute these values without explicitly representing $Q_M^\pi$, and each value can be easily estimated by averaging over multiple Monte-Carlo rollouts [50], i.e., $Q_M^\pi(s,a) = \mathbb{E}_\pi[\sum_{t=0}^\infty \gamma^t r_t | s_0 = s, a_0 = a]$ can be approximated by rolling out multiple trajectories starting from $(s,a)$ and taking actions according to $\pi$. For the model-based estimators proposed in Section 4, we need access to quantities in the form of $(\mathcal{T}_{M_j}^\pi Q_{M_i}^\pi)(s,a)$ (see Appendix B). This value can also be obtained by Monte-Carlo simulation: (1) start in $(s,a)$ and simulate one step in $M_j$, then (2) switch to $M_i$, simulate from step 2 onwards and rollout the rest of the trajectory.

## 5.2 Computational Efficiency

Despite not involving neural-net optimization, the experiment can still be computationally intensive due to rolling out a large number of trajectories. In our code, we incorporate the following measures to reduce the computational cost:

**Q-caching.** The most intensive computation is to roll-out Monte-Carlo trajectories for Q-value estimation; the cost of running the actual selection algorithms is often much lower and negligible. Hence, we generate these MC Q-estimates and save them to files, and retrieve them during selection. This makes it efficient to experiment with new selection algorithms or add extra baselines, and also enables fast experiment that involves a subset of the candidate models (see Section 6.2).

**Bootstrapping.** To account for the randomness of $\mathcal{D}$, we use bootstrapping to sample (with replacement) multiple datasets, and report an algorithm's mean performance across these bootstrapped samples with 95% confidence intervals. Using bootstrapping maximally reuses the cached Q-values and avoids the high computational costs of sampling multiple datasets and performing Q-caching in each of them, which is unavoidable if we were to repeat each experiment verbatim multiple times.

# 6 Exemplification of the Protocol

In this section we instantiate our protocol in the Gym Hopper environment to demonstrate its utility, while also providing preliminary empirical results for our algorithms. Our code is available at `https://github.com/Coder-PAI/2025_neurips_model_selection_rl.git`.

## 6.1 Experiment Setup and Main Results

Our experiments will be based on the *Hopper-v4* environment [9]. To create a variety of environments, we add different levels of stochastic noise in the transitions and change the gravity constant (see

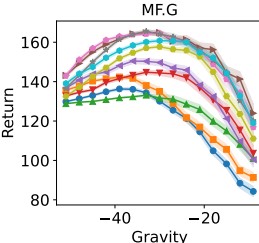

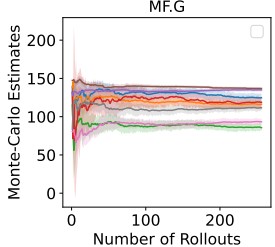

Figure 2: **Left:** $J_M(\pi)$ in $M \in \mathcal{M}_{\mathbf{g}}$ (cf. Section 6.1) for different target policies. **Right:** Convergence of Monte-Carlo estimates of $J(\pi)$. Each curve corresponds to a target policy.

Appendix D.1). Each environment is then parameterized by the gravity constant **g** and noise level **n**. We consider arrays of such environments as the set of candidate simulator $\mathcal{M}$: in most of our results, we consider a "gravity grid" (denoted using **MF.G** in the figures) $\mathcal{M}_{\mathbf{g}} := \{M_{\mathbf{g}}^0 \ldots, M_{\mathbf{g}}^{14}\}$ (fixed noise level, varying gravity constants from $-51$ to $-9$) and a "noise grid" (**MF.N**) $\mathcal{M}_{\mathbf{n}} := \{M_{\mathbf{n}}^0 \ldots, M_{\mathbf{n}}^{14}\}$ (fixed gravity constant, varying noise level from $10$ to $100$). Each array contains 15 environments, though some subsequent results may only involve a subset of them (Section 6.2). Some of these simulators will also be treated as groundtruth environment $M^{\star}$, which determines the groundtruth performance of target policies and produces the offline dataset $\mathcal{D}$.

**Behavior and Target Policies.** We create 15 target policies by running DDPG [37] in one of the environments and take checkpoints. For each $M^{\star}$, the behavior policy is the randomized version of one of the target policies; see Appendix D.2 for details. A dataset is collected by sampling trajectories until $n = 3200$ transition tuples are obtained. As a sanity check, we plot $J_M(\pi)$ for $\pi \in \Pi_{\mathbf{g}}$ and $M \in \mathcal{M}_{\mathbf{g}}$ in Figure 2. As can be shown in the figure, the target policies have different performances, and also vary in a nontrivial manner w.r.t. the gravity constant **g**. It is important to perform such a sanity check to avoid degenerate settings, such as $J_M(\pi)$ varies little across $M \in \mathcal{M}$ (then even a random selection will be accurate) or across $\pi \in \Pi$.

**Number of Rollouts.** We then decide the two important parameters for estimating the Q-value, the number of Monte-Carlo rollouts $l$ and the horizon (i.e., trajectory length) $H$. For horizon, we set $H = 1024$ which is substantially longer than typically observed trajectories from the target policies. For $l$, we plot the convergence of $J_M(\pi)$ estimation and choose $l = 128$ accordingly (see Figure 2R).

**Compared Methods.** We compare our methods with baselines, including TD-square (Eq.(2)), naïve model-based (Eq.(12)), BVFT [64], and "average Bellman error" $|E_{\mathcal{D}}[Q_i(s, a) - r - \gamma Q_i(s', \pi)]|$ [25], which can be viewed as our LSTD-Tournament but with a trivial constant discriminator. The model-based methods in Section 4 require MC rollouts for $\{T_{M_j}^{\pi} Q_{M_i}^{\pi} : i, j \in [m]\}$, which requires $O(m^2)$ computational complexity. Therefore, we first compare other selectors (mostly model-free) in Figure 3 with $m = 15$; the relatively large number of candidate simulators will also enable the

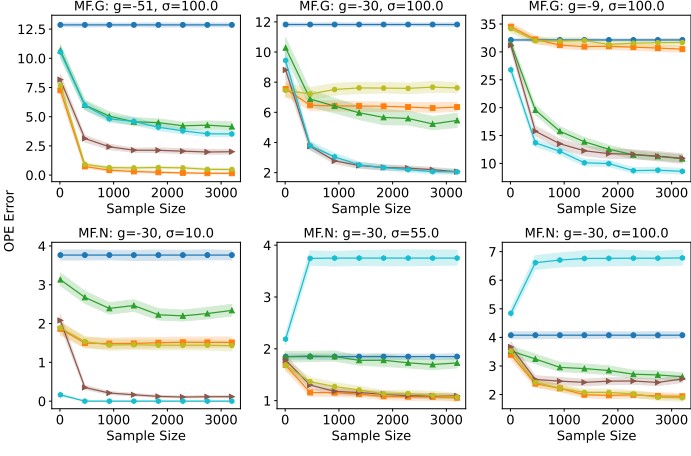

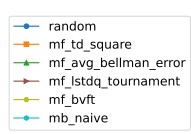

Figure 3: Main results for comparing *model-free* selectors in the gravity grid (**MF.G**; top row) and the noise grid (**MF.N**; bottom row). Each plot corresponds to a different $M^{\star}$ as indicated in the plot title. "mb_naive" is model-based but still included since it does not require Bellman operator rollouts.

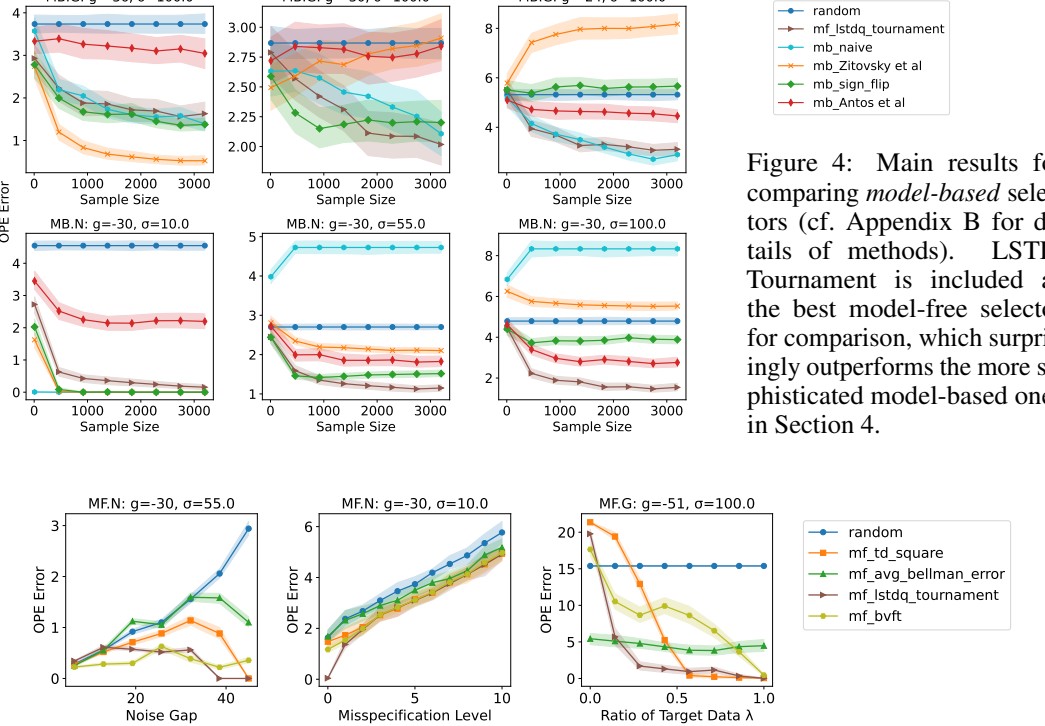

Figure 4: Main results for comparing *model-based* selectors (cf. Appendix B for details of methods). LSTD-Tournament is included as the best model-free selector for comparison, which surprisingly outperforms the more sophisticated model-based ones in Section 4.

Figure 5: **Left:** OPE error vs. simulator gaps. **Middle:** OPE error vs. misspecification. **Right:** OPE error vs. data coverage.

later subgrid studies in Section 6.2. We then perform a separate experiment with $m = 5$ for the model-based selectors (Figure 4).

**Main Results.**    Figure 3 shows the main model-free results. Our LSTD-Tournament method demonstrates strong and reliable performance. Note that while some methods sometimes outperform it, they suffer catastrophic performances when the true environment changes. For example, the naïve model-based method performs poorly in high-noise environment ($\sigma = 55.0$ and $100.0$) when the candidate models have varying degrees of stochasticity (**MF.N**), as predicted by theory (Section 4). BVFT's performance mostly coincides with TD-sq, which is a possible degeneration predicted by [64]. This is particularly plausible when the number of data points $n$ is not large enough to allow for meaningful discretization and partition of the state space required by the method.

Figure 4 shows the result on smaller candidate sets (**MB.G** and **MB.N**; see Appendix D.2), where we implement the three model-based selectors in Section 4 whose computational complexities grow quadratically with $|\mathcal{M}|$. Our expectation was that (1) these algorithms should address the double-sampling issue and will outperform naïve model-based when the latter fails catastrophically, and (2) by having access to more information, model-based should outperform model-free algorithms under realizability. While the first prediction is largely verified, we are surprised to find that the second prediction went wrong, and our LSTD-Tournament method is more robust and generally outperforms the more complicated model-based selectors.

## 6.2   Subgrid Studies: Gaps and Misspecifications

We now demonstrate how to extract additional insights from the Q-values cached earlier. Due to space limit we are only able to show representative results in Figure 5, and more comprehensive results can be found in Appendix E.

**Gaps.**   We investigate an intellectually interesting question: is the selection problem easier if the candidate simulators are very similar to each other, or when they are very different? We argue that the answer is **neither**, and an intermediate difference (or *gap*) is the most challenging: if the

simulators are too similar, their $J_M(\pi)$ predictions will all be close to $J_{M^\star}(\pi)$ since $M \approx M^\star$, and any selection algorithm will perform well; if the simulators are too dissimilar, it should be easy to tell them apart, which also makes the task easy.

To empirically test this, we let $M^\star = M_{\mathbf{n}}^7$ and run the experiments with different 3-subsets of $\mathcal{M}_{\mathbf{n}}$, including $\{6, 7, 8\}$ (least gap), $\{5, 7, 9\}, \ldots, \{0, 7, 14\}$ (largest gap). Since the needed Q-values have already been cached in the main experiments, we can skip caching and directly run the selection algorithms. We plot the prediction error as a function of gap size in Figure 5L, and observe the down-U curves (except for trivial methods such as random) as expected from earlier intuition.

**Misspecification.** Similarly, we study the effect of misspecification, i.e., $M^\star \notin \mathcal{M}$. For example, we can take $M^\star = M_\sigma^0$, and consider different subsets of $\mathcal{M}_{\mathbf{n}}$: 0–4 (realizable), 1–5 (low misspecification), $\ldots$, 10–14 (high misspecification). Figure 5M plots OPE error vs. misspecification level, where we expect to observe potential difference in the sensitivity to misspecification. The actual result is not that interesting given similar increasing trends for all methods.

## 6.3   Data Coverage

In the previous subsection, we have seen how multiple experiment units that only differ in $\mathcal{M}$ can provide useful insights. Here we show that we can also probe the methods' sensitivity to data coverage by looking at experiment units that only differ in the dataset $\mathcal{D}$. In Figure 5R, we take a previous experiment setting ($\mathcal{M}_{\mathbf{g}}$) and isolate a particular target policy $\pi$; then, we create two datasets: (1) $\mathcal{D}_\pi$ sampled using $\pi$; (2) $\mathcal{D}_{\text{off}}$ sampled using a policy that is created to be very different from the target policies and offer very little coverage (see Appendix D.2). Then, we run the algorithm with $\lambda$ fraction of data from $\mathcal{D}_\pi$ combined with $(1 - \lambda)$ from $\mathcal{D}_{\text{off}}$; as predicted by theory, most methods perform better with better coverage (large $\lambda$), and performance degrades as $\lambda$ goes to 0.

# 7   Limitations, Future Directions, and Conclusion

We conclude the paper with a discussion of the limitations of our work and potential future directions.

**Realism of Candidate Q-functions.** While our proposed experimental protocol offers significant advantages in controllability and stability (c.f. Footnote 2), a key limitation lies in the realism of the candidate Q-functions it generates. In practice, these functions will likely come from learning algorithms (e.g., TD/FQE), whose errors may differ from the structured errors induced by our protocol (e.g., varying gravity *uniformly* across states). Varying environment parameters in more complex, state-dependent ways could potentially generate more realistic error patterns, presenting an interesting avenue for future investigation. That said, the prior protocol using FQE may also face realism challenges, just in a different way: in practice, we carefully tune the optimization of OPE algorithms for each target policy to produce reasonable candidates, which can be infeasible in empirical benchmarking given the sheer number of policies we are working with, resulting in poorer candidates. Neither approach perfectly mirrors reality. In addition, our protocol offers controllable quality that enables targeted studies (like the gap experiments in Section 6.2), so for more comprehensive empirical studies, comparing results from both protocols is advisable.

**Fundamental Theory of LSTDQ.** As discussed in Appendix C.5, our theoretical analysis of LSTD-Tournament reveals limitations and open questions for the coverage assumption made by LSTDQ ($\sigma_{\min}(A)$ in Theorem 1), which is standard in the literature and inherited by LSTD-Tournament. Such conditions differ significantly from the more standard concentrability coefficients (e.g., $\mathcal{C}_\infty^\pi$ in Theorems 3 and 4), and the lack of satisfactory understanding of such a classic algorithm calls for future investigation. We hope to study this question in the future, and any progress would directly improve the guarantees and understanding of LSTD-Tournament.

As another potential future direction, LSTDQ can be viewed as an application of *instrumental-variable* regression in value-function estimation [8, 11], where the left $\phi$ in the moments of Eq.(3) plays the role of an instrument. While such a choice is standard and natural, it will be interesting to explore alternative choices of instruments and examine if they can lead to improved theoretical guarantees and practical performance in the model-selection problem.

## Acknowledgements

Nan Jiang acknowledges funding support from NSF CNS-2112471, NSF CAREER IIS-2141781, Google Scholar Award, and Sloan Fellowship.

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

# A    Other Related Works

**Adaptivity Guarantees for Offline RL Model Selection.**   Our theoretical guarantees state that the selected candidate function/model provides a $J(\pi)$ estimate that is close to the groundtruth, as if the algorithm knew which candidate is correct. This can be viewed as a form of adaptivity guarantees, which is the goal of several theoretical works on model selection. Su et al. [51] and Udagawa et al. [56] study adaptive model selection for OPE, but their approaches crucially rely on the importance sampling estimator, which we do not consider due to the focus on long-horizon tasks. Lee et al. [33] (who build on and improve upon the earlier works of Farahmand and Szepesvári [15] and Jiang et al. [24]) study model-selection of value functions, with a focus on the double-sampling issue and its relationship with Bellman completeness; these are also the very core consideration in our theoretical derivations. However, their approach treats the OPE instances (such as FQE) in a less black-box manner, and the goal is to select the "right" function *class* for the OPE algorithm (instead of directly selecting a final output function) from a nested series of classes, one of which is assumed to be Bellman-complete [6, 10] and have low generalization errors.[5] This makes their approach less widely applicable than ours, and we cannot empirically evaluate their algorithm in our protocol as a consequence. Nevertheless, their results provide an interesting alternative and valuable insights to the model-selection problem, which are "complementary" to the BVFT line of work [62, 64] that we further develop. Similarly, Miyaguchi [41] considers the selection of Bellman operators, which can be instantiated as regression using different function classes, making their approach similar to Lee et al. [33] in spirit.

**Other Works on Model Selection.**   Apart from the above works and those already discussed in the main text, most related works are not concerned about new selection algorithms with theoretical guarantees or experiment protocol for OPE model selection (see Voloshin et al. [60], Kiyohara et al. [26] for experiment protocol and benchmarks of OPE itself), so their focus is different and often provides insights complementary to our work. For example, Nie et al. [46] discuss data splitting in offline model selection; this is a question we avoid by assuming a fixed holdout dataset for OPE model selection. Tang and Wiens [54] compare importance sampling methods and FQE and conclude that FQE is more effective (which echos with Paine et al. [47]), but does not provide provable methods for selecting the hyperparameters of FQE, especially the choice of function approximation. Kumar et al. [28] provide a pipeline for offline RL that includes hyperparameter selection as a component, but the recommendations are heuristics specialized to algorithms such as CQL [27].

**Debate on Bellman Error as a Proxy.**   Most of the selectors we consider estimate some variants of the Bellman error. Regarding this, Fujimoto et al. [16] challenge the idea of using Bellman errors for model selection due to their surrogacy and poor correlation with the actual objective. Despite the valid criticisms, there are no clear alternatives that address the pain points of Bellman errors, and the poor performance is often due to lack of data coverage, which makes the task fundamentally difficult for any algorithm. We still believe that Bellman-error-like objectives (defined in a broad sense, which includes our LSTD-Tournament) are promising for model selection, and the improvement on OPE error (which is what we report in experiments) is the right goal to pursue instead of correlation (which we know could be poor due to the surrogate nature of Bellman errors).

**Data Coverage in Model Selection.**   As mentioned above and demonstrated in our experiments, the lack of data coverage is a key factor that determines the difficulty of the selection tasks. Lee et al. [32] propose feature selection algorithms for offline contextual bandits that account for the different coverage effects of candidate features, but it is unclear how to extend the ideas to MDPs. On a related note, ideas from offline RL training, such as version-space-based pessimism [63], can also be incorporated in our method. This will unlikely improve the accuracy of OPE itself, but may be helpful if we measure performance by how OPE can eventually lead to successful selection of performant policies, which we leave for future investigation.

**Experimental Protocol.**   The closest to our experimental protocol is the work of Sajed et al. [50] who also uses Monte-Carlo rollouts to produce unbiased estimates of value functions on individual states. Other than this point, their work is largely orthogonal to ours as they focus on establishing high-confidence bounds for the estimated value, assuming the simulator is the groundtruth environment,

---

[5]Given the relationship between Bellman-completeness and bisimulation abstractions [10, Proposition 9], their setting is a natural generalization of Jiang et al. [24]'s setting of selecting a bisimulation from a nested series of state abstractions.

whereas we draw Monte-Carlo trajectories from different simulators to facilitate the model-selection problem.

# B  Details of Model-based Selectors

Here we expand Section 4 on the model-based setting, i.e., choosing a model from $\{M_i\}_{i\in[m]}$. This is a practical scenario when we have structural knowledge of the system dynamics and can build reasonable simulators, but simulators of complex real-world systems will likely have many design choices and knobs that cannot be set from prior knowledge alone. In some sense, the task is not very different from system identification in control and model learning in model-based RL, except that (1) we focus on a finite and small number of plausible models, instead of a rich and continuous hypothesis class, and (2) the ultimate goal is to perform accurate OPE, and learning the model is only an intermediate step.

**Existing Methods.**   Given the close relation to model learning, a natural approach is to simply minimize the model prediction loss [21]: a candidate model $M$ is scored by

$$\mathbb{E}_{(s,a,s')\sim\mu,\tilde{s}\sim P(\cdot|s,a)}[d(s',\tilde{s})], \tag{12}$$

where $s'$ is in the data and generated according to the real dynamics $P^\star(\cdot|s,a)$, and $\tilde{s}$ is generated from the candidate model $M$'s dynamics $P$. $d(\cdot,\cdot)$ is a distance metric that measures the difference between states.

Despite its wide use and simplicity [17, 45, 18], the method has major caveats: first, the distance metric $d(\cdot,\cdot)$ is a design choice. When the state is represented as a real-valued vector, it is natural to use the $\ell_2$ distance as $d(\cdot,\cdot)$, which changes if we simply normalize/rescale the coordinates. Second, the metric is biased for stochastic environments as discussed in prior works [21, 59], which we also demonstrate in the experiment section (Section 6); essentially this is a version of the double-sampling issue but for the model-based setting [5]. As a minimal counterexample, suppose $\mathcal{S}\subset\mathbb{R}^2$ and we focus on the transition distribution from a particular $(s,a)$ pair where groundtruth is uniform over 4 points $(\pm 1,\pm 1)$. The expected loss of the groundtruth model itself, as in Eq.(12), is $1+\sqrt{2}/2$; this is higher than a wrong model that always predicts 0 deterministically, which yields a loss of $\sqrt{2}$.

There are alternative methods that address these issues. For example, in the theoretical literature, MLE losses are commonly used, i.e., $\mathbb{E}_\mu[\log P(s'|s,a)]$ [1, 58, 39], which avoids $d(\cdot,\cdot)$ and works properly for stochastic MDPs by effectively measuring the KL divergence between $P^\star(\cdot|s,a)$ and $P(\cdot|s,a)$. Unfortunately, most complex simulators do not provide explicit probabilities $P(s'|s,a)$, making it difficult to use in practice. Moreover, when the support of $P^\star(\cdot|s,a)$ is not fully covered by $P(\cdot|s,a)$, the loss can become degenerate.

To address these issues, we propose to estimate the Bellman error $\mathbb{E}_\mu[(Q_i-\mathcal{T}^\pi Q_i)^2]$, where $Q_i := Q_{M_i}^\pi$. As discussed earlier, this objective suffers the double-sampling issue in the model-free setting, which we show can be addressed when we have access to candidate models $\{M_1,\ldots,M_m\}$ that contains the true dynamics $M^\star$. Moreover, the Bellman error $|Q_{M_i}^\pi(s,a)-(\mathcal{T}^\pi Q_{M_i}^\pi)(s,a)| =$

$$\gamma|\mathbb{E}_{s'\sim P^\star(\cdot|s,a)}[Q_i(s',\pi)]-\mathbb{E}_{s'\sim P_i(\cdot|s,a)}[Q_i(s',\pi)]|,$$

which can be viewed as an IPM loss [43] that measures the divergence between $P^\star(\cdot|s,a)$ and $P(\cdot|s,a)$ under $Q_i(\cdot,\pi)$ as a discriminator. IPM is also a popular choice of model learning objective in theory [52, 61], and the Bellman error provides a natural discriminator relevant for the ultimate task of interest, namely OPE.

## B.1  Regression-based Selector

Recall that the difficulty in estimating the Bellman error $\mathbb{E}_\mu[(Q_i-\mathcal{T}^\pi Q_i)^2]$ is the uncertainty in $\mathcal{T}^\pi$. To overcome this, we leverage the following observation from Antos et al. [6], where for any $f:\mathcal{S}\times\mathcal{A}\to\mathbb{R}$,

$$\mathcal{T}^\pi f \in \underset{g:\mathcal{S}\times\mathcal{A}\to\mathbb{R}}{\arg\min}\ \mathbb{E}_\mu[(g(s,a)-r-\gamma f(s',\pi))^2], \tag{13}$$

which shows that we can estimate $\mathcal{T}^\pi Q_i$ by solving a sample-based version of the above regression problem with $f=Q_i$. Statistically, however, we cannot afford to minimize the objective over

all possible functions $g$; we can only search over a limited set $\mathcal{G}_i$ that ideally captures the target $\mathcal{T}^\pi Q_i$. Crucially, in the model-based setting we can generate such a set directly from the candidates $\{M_i\}_{i\in[m]}$:

**Proposition 1.** *Let* $\mathcal{G}_i := \{\mathcal{T}^\pi_{M_j} Q_i : j \in [m]\}$. *Then if* $M^\star \in \{M_i\}_{i\in[m]}$, *it follows that* $\mathcal{T}^\pi Q_i = \mathcal{T}^\pi_{M^\star} Q_i \in \mathcal{G}_i$.

The constructed $\mathcal{G}_i$ ensures that regression is statistically tractable given its small cardinality, $|\mathcal{G}_i| = m$. To select $Q_i$, we choose $Q_i$ with the smallest loss defined as follows:

1. $\widehat{g}_i := \arg\min_{g\in\mathcal{G}_i} \mathbb{E}_\mathcal{D}[(g(s,a) - r - \gamma Q_i(s',\pi))^2]$.
2. The loss of $Q_i$ is $\mathbb{E}_\mathcal{D}[(\widehat{g}_i(s,a) - Q_i(s,a))^2]$.

The 2nd step follows from Zitovsky et al. [65]. Alternatively, we can also use the min value of Eq.(13) (instead of the argmin function) to correct for the bias in TD-squared (Eq.(2)) [6]; see Liu et al. [40] for another related variant. These approaches share similar theoretical guarantees under standard analyses [62, 63], and we only state the guarantee for the Zitovsky et al. [65] version below, but will include both in the experiments ("mb_Zitovsky et al." and "mb_Antos et al" in Figure 4).

**Theorem 3.** *Let* $\mathcal{C}^\pi := \mathbb{E}_\pi\left[\frac{d^\pi(s,a)}{\mu(s,a)}\right]$. *For* $Q_{\widehat{i}}$ *that minimizes* $\mathbb{E}_\mathcal{D}[(\widehat{g}_i(s,a) - Q_i(s,a))^2]$ *we have w.p.* $\geq 1 - \delta$,

$$J(\pi) - \mathbb{E}_{d_0}\left[Q_{\widehat{i}}(s,\pi)\right] \leq \frac{V_{\max}}{1-\gamma}\sqrt{\frac{152 \cdot \mathcal{C}^\pi \cdot \log\left(\frac{4m}{\delta}\right)}{n}}. \tag{14}$$

### B.2 Sign-flip Average Bellman Error

We now present another selector that leverages the information of $\mathcal{G}_i = \{\mathcal{T}^\pi_{M_j} : j \in [m]\}$ in a different manner. Instead of measuring the squared Bellman error, we can also measure the absolute error, which can be written as (some $(s,a)$ argument to functions are omitted):

$$\begin{aligned}
&\mathbb{E}_\mu[|Q_i - \mathcal{T}^\pi_{M^\star} Q_i|] \\
&= \mathbb{E}_\mu[\text{sgn}(Q_i(s,a) - (\mathcal{T}^\pi Q_i)(s,a))(Q_i - \mathcal{T}^\pi Q_i)] \\
&= \mathbb{E}_\mu[\text{sgn}(Q_i - \mathcal{T}^\pi Q_i)(Q_i(s,a) - r - \gamma Q_i(s',\pi))] \\
&\leq \max_{g\in\mathcal{G}_i} \mathbb{E}_\mu[\text{sgn}(Q_i - g)(Q_i(s,a) - r - \gamma Q_i(s',\pi))].
\end{aligned} \tag{15}$$

Here, the $\mathcal{G}_i$ from Proposition 1 induces a set of sign functions $\text{sgn}(Q_i - g)$, which includes $Q_i - \mathcal{T}^\pi Q_i$, that will negate any negative TD errors. The guarantee is as follows:

**Theorem 4.** *Let* $Q_{\widehat{i}}$ *be the minimizer of the empirical estimate of Eq.(15), and* $\mathcal{C}^\pi_\infty := \max_{s,a} \frac{d^\pi(s,a)}{\mu(s,a)}$. *W.p.* $\geq 1 - \delta$,

$$J(\pi) - \mathbb{E}_{d_0}\left[Q_{\widehat{i}}(s,\pi)\right] \leq 4 \cdot \mathcal{C}^\pi_\infty \cdot V_{\max}\sqrt{\frac{\log(2m/\delta)}{n}}.$$

## C  Proofs

### C.1  Proof of Theorem 1

*Proof.* Define the following loss vectors,

$$\begin{aligned}
\ell(\theta) &:= A\theta - b \in \mathbb{R}^d, \\
\widehat{\ell}(\theta) &:= \widehat{A}\theta - \widehat{b} \in \mathbb{R}^d
\end{aligned}$$

and recall that we select as the estimator

$$\widehat{\theta} := \arg\min_{\theta\in\Theta} \|\widehat{\ell}(\theta)\|_\infty.$$

Since $\theta^\star = A^{-1}b$, we can write the desired bound as a function of $\ell(\theta)$ as follows,

$$
\begin{aligned}
\|Q^\pi(\cdot) - \phi^\top(\cdot)\widehat{\theta}\|_\infty &= \|\phi^\top(\cdot)(\widehat{\theta} - \theta^\star)\|_\infty \\
&= \|\phi^\top(\cdot)A^{-1}(A\widehat{\theta} - b)\|_\infty \\
&= \|\phi^\top(\cdot)A^{-1}\ell(\widehat{\theta})\|_\infty \\
&= \max_{s,a}|\phi^\top(s,a)A^{-1}\ell(\widehat{\theta})| \\
&\leq \left(\max_{s,a}\|\phi^\top(s,a)\|_2\right)\cdot\|A^{-1}\|_2\cdot\|\ell(\widehat{\theta})\|_2 \\
&\leq \sqrt{d}B_\phi\cdot\|A^{-1}\|_2\cdot\|\ell(\widehat{\theta})\|_\infty
\end{aligned}
$$

Next, we control the $\ell(\widehat{\theta})$ term. In the sequel we will establish via concentration that

$$
\|\ell(\theta) - \widehat{\ell}(\theta)\|_\infty \leq \varepsilon_{\mathsf{stat}} := 3\cdot\max\{R_{\max}, B_\phi\}^2\cdot\sqrt{\frac{\log(2d|\Theta|\delta^{-1})}{n}}, \quad \forall\theta\in\Theta. \qquad (16)
$$

Then, we have that

$$
\begin{aligned}
\|\ell(\widehat{\theta})\|_\infty &\leq \|\widehat{\ell}(\widehat{\theta})\|_\infty + \varepsilon_{\mathsf{stat}} \\
&\leq \|\widehat{\ell}(\theta^\star)\|_\infty + \varepsilon_{\mathsf{stat}} \\
&\leq \|\ell(\theta^\star)\|_\infty + 2\cdot\varepsilon_{\mathsf{stat}} \\
&= 2\cdot\varepsilon_{\mathsf{stat}},
\end{aligned}
$$

where we recall that $\widehat{\theta} = \arg\min_{\theta\in\Theta}\|\widehat{\ell}(\theta)\|_\infty$ in the second inequality, and that $A\theta^\star = b$ in the last line. Combining the above, we obtain

$$
\begin{aligned}
\|Q^\pi - \phi^\top\widehat{\theta}\|_\infty &\leq 2\sqrt{d}B_\phi\cdot\|A^{-1}\|_2\cdot\varepsilon_{\mathsf{stat}} \\
&= 6\sqrt{d}\cdot\|A^{-1}\|_2\cdot\max\{R_{\max}, B_\phi\}^2\cdot\sqrt{\frac{\log(2d|\Theta|\delta^{-1})}{n}},
\end{aligned}
$$

as desired. We now establish the concentration result of Equation (16).

**Concentration results.** For $j\in[d]$, let $\phi_j(s,a)\in\mathbb{R}$ refer to the $j$'th entry of the vector. For any $(s,a,s')$ and $\theta$, define

$$
B^\pi(s,a,s';\theta) := \phi^\top(s,a)\theta - \gamma\phi^\top(s',\pi)\theta - r(s,a)
$$

Recall that $\|\phi(s,a)\|_2 \leq B_\phi$ for all $(s,a)$ and that $\|\theta\|_2 \leq B_\Theta$ for all $\theta\in\Theta$. We have that, for all $j\in[d]$, $\theta\in\Theta$, and $s,a\in\mathcal{S}\times\mathcal{A}$ we have that $\phi_j(s,a)B^\pi(s,a,s';\theta)$ is bounded, since:

$$
\begin{aligned}
&\phi_j(s,a)\big(\phi^\top(s,a)\theta - \gamma\phi^\top(s',\pi)\theta - r(s,a)\big) \\
&\leq \|\phi(s,a)\|_\infty(\|\phi(s,a)\|_2\|\theta\|_2 + \gamma\|\phi(s',\pi)\|_2\|\theta\|_2 + R_{\max}) \\
&\leq \max_{s,a}\|\phi(s,a)\|_2\left(\max_{s,a}\|\phi(s,a)\|_2\|\theta\|_2 + \gamma\max_{s,a}\|\phi(s,a)\|_2\|\theta\|_2 + R_{\max}\right) \\
&\leq (1+\gamma)B_\phi^2 + R_{\max}B_\phi \\
&\leq 3\max\{B_\phi, R_{\max}\}^2.
\end{aligned}
$$

Thus, from Hoeffding's inequality and a union bound, we have that for all $j\in[d]$ and $\theta\in\Theta$:

$$
\left|\mathbb{E}_\mu\big[\phi^j(s,a)B^\pi(s,a,s';\theta)\big] - \widehat{\mathbb{E}}_\mu\big[\phi^j(s,a)B^\pi(s,a,s';\theta)\big]\right| \leq 3\max\{B_\phi, R_{\max}\}^2\sqrt{\frac{2\log(d|\Theta|\delta^{-1})}{n}} = \varepsilon_{\mathsf{stat}},
$$

with probability at least $1-\delta$. As a result, we can write

$$
\begin{aligned}
\left\|\ell(\theta) - \widehat{\ell}(\theta)\right\|_\infty &= \left\|\mathbb{E}_\mu\big[\phi(s,a)\big(\phi^\top(s,a)\theta - \gamma\phi^\top(s',\pi)\theta - r(s,a)\big)\big] - \widehat{\mathbb{E}}_\mu\big[\phi(s,a)\big(\phi^\top(s,a)\theta - \gamma\phi^\top(s',\pi)\theta - r(s,a)\big)\big]\right\|_\infty \\
&= \left\|\mathbb{E}_\mu[\phi(s,a)B^\pi(s,a,s';\theta)] - \widehat{\mathbb{E}}_\mu[\phi(s,a)B^\pi(s,a,s';\theta)]\right\|_\infty \\
&\leq \|\mathbf{1}\cdot\varepsilon_{\mathsf{stat}}\|_\infty \\
&\leq \varepsilon_{\mathsf{stat}}.
\end{aligned}
$$

This concludes the proof. $\qquad\square$

## C.2 Proof of Theorem 2

*Proof.* We first note that the proposed algorithm is equivalent to the following tournament procedure:

- $\forall i \in [m], j \neq i$ :
  - Define $\phi_{i,j}(s,a) := [Q_i(s,a), Q_j(s,a)]^\top$ and associated $\hat{A}_{i,j}$ matrix and $\hat{b}_{i,j}$ vector (Eq. 4)
  - Define $\hat{\ell}_{i,j} = \hat{A}_{i,j} e_1 - \hat{b}_{i,j} \in \mathbb{R}^2$
- Pick $\arg\min_{i \in [m]} \max_{j \neq i} \|\hat{\ell}_{i,j}\|_\infty$

Let $i^\star \in [m]$ denote the index of $Q^\pi$ in the enumeration of $\mathcal{Q}$. We start with the upper bound

$$|J_{M^\star}(\pi) - \mathbb{E}_{s \sim d_0}[Q_{\hat{i}}(s, \pi)]| = |\mathbb{E}_{s \sim d_0}[Q^\pi(s, \pi)] - \mathbb{E}_{s \sim d_0}[Q_{\hat{i}}(s, \pi)]| \leq \|Q^\pi(\cdot) - Q_{\hat{i}}(\cdot)\|_\infty.$$

Let $\ell_{i,j} := A_{i,j} e_1 - b_{i,j}$ denote the population loss. We recall the concentration result from Equation (16), which, for any fixed $i$ and $j$, implies:

$$\|\ell_{i,j} - \widehat{\ell}_{i,j}\|_\infty \leq \varepsilon_{\mathsf{stat}} = 3 \cdot \max\{B_\phi, R_{\max}\}^2 \cdot \sqrt{\frac{\log(2d\delta^{-1})}{n}},$$

with probability at least $1 - \delta$. This further implies $|\|\ell_{i,j}\|_\infty - \|\hat{\ell}_{i,j}\|_\infty| \leq \varepsilon_{\mathsf{stat}}$. Taking a union bound over all $(i,j)$ where either $i$ or $j$ equal $i^\star$, this implies that

$$\|\ell_{i,j} - \widehat{\ell}_{i,j}\|_\infty \leq \varepsilon_{\mathsf{stat}} = 3 \cdot \max\{B_\phi, R_{\max}\}^2 \cdot \sqrt{\frac{\log(4dm\delta^{-1})}{n}} \quad \forall (i,j) \in ([m] \times \{i^\star\}) \cup (\{i^\star\} \times [m])$$

If $\hat{i} = i^\star$ then we are done. If not, then there exists a comparison in the tournament where $i = \hat{i}$ and $j = i^\star$. For these features $\phi_{\hat{i},i^\star}(s,a) = [Q_{\hat{i}}(s,a), Q_{i^\star}(s,a)]^\top$, we have:

$$
\begin{aligned}
\|Q^\pi(\cdot) - Q_{\hat{i}}(\cdot)\|_\infty &= \|\phi_{\hat{i},i^\star}^\top (e_2 - e_1)\|_\infty \\
&= \|\phi_{\hat{i},i^\star}^\top A_{\hat{i},i^\star}^{-1} A_{\hat{i},i^\star} (e_2 - e_1)\|_\infty \\
&= \max_{s,a} |\phi_{\hat{i},i^\star}^\top (s,a) A_{\hat{i},i^\star}^{-1} A_{\hat{i},i^\star} (e_2 - e_1)| \\
&\leq \left( \max_{s,a} \|\phi_{\hat{i},i^\star}(s,a)\|_2 \right) \|A_{\hat{i},i^\star}^{-1}\|_2 \|A_{\hat{i},i^\star}(e_2 - e_1)\|_2 \\
&\leq \sqrt{d} \left( \max_{s,a} \|\phi_{\hat{i},i^\star}(s,a)\|_2 \right) \|A_{\hat{i},i^\star}^{-1}\|_2 \|A_{\hat{i},i^\star}(e_2 - e_1)\|_\infty \\
&= \sqrt{d} \left( \max_{s,a} \|\phi_{\hat{i},i^\star}(s,a)\|_2 \right) \|A_{\hat{i},i^\star}^{-1}\|_2 \|A_{\hat{i},i^\star} e_1 - b_{\hat{i},i^\star}\|_\infty \\
&= \sqrt{d} \left( \max_{s,a} \|\phi_{\hat{i},i^\star}(s,a)\|_2 \right) \|A_{\hat{i},i^\star}^{-1}\|_2 \|\ell_{\hat{i},i^\star}\|_\infty \\
&\leq \sqrt{d} \left( \max_{s,a} \|\phi_{\hat{i},i^\star}(s,a)\|_2 \right) \|A_{\hat{i},i^\star}^{-1}\|_2 \left( \|\hat{\ell}_{\hat{i},i^\star}\|_\infty + \varepsilon_{\mathsf{stat}} \right) \\
&\leq \sqrt{d} \left( \max_{s,a} \|\phi_{\hat{i},i^\star}(s,a)\|_2 \right) \|A_{\hat{i},i^\star}^{-1}\|_2 \left( \max_{j \in [m] \setminus \{i^\star\}} \|\hat{\ell}_{\hat{i},j}\|_\infty + \varepsilon_{\mathsf{stat}} \right) \\
&\leq \sqrt{d} \left( \max_{s,a} \|\phi_{\hat{i},i^\star}(s,a)\|_2 \right) \|A_{\hat{i},i^\star}^{-1}\|_2 \left( \max_{j \in [m] \setminus \{i^\star\}} \|\hat{\ell}_{i^\star,j}\|_\infty + \varepsilon_{\mathsf{stat}} \right) \\
&\leq \sqrt{d} \left( \max_{s,a} \|\phi_{\hat{i},i^\star}(s,a)\|_2 \right) \|A_{\hat{i},i^\star}^{-1}\|_2 \left( \max_{j \in [m] \setminus \{i^\star\}} \|\ell_{i^\star,j}\|_\infty + 2\varepsilon_{\mathsf{stat}} \right) \\
&\leq 2\sqrt{d} \left( \max_{s,a} \|\phi_{\hat{i},i^\star}(s,a)\|_2 \right) \|A_{\hat{i},i^\star}^{-1}\|_2 \varepsilon_{\mathsf{stat}}. \\
&\leq 2\sqrt{d} \left( \max_{s,a} \|\phi_{\hat{i},i^\star}(s,a)\|_2 \right) \max_{i \in [m] \setminus \{i^\star\}} \frac{1}{\sigma_{\min}(A_{i,i^\star})} \varepsilon_{\mathsf{stat}}.
\end{aligned}
$$

To conclude, we note that $d = 2$ in our application and that $\max_{s,a}\|\phi_{i,j}(s,a)\|_2^2 = Q_i^2(s,a) + Q_j^2(s,a) \le 2V_{\max}^2$. Plugging in the value for $\varepsilon_{\mathsf{stat}}$, this gives a final bound of

$$|J_{M^\star}(\pi) - \mathbb{E}_{s \sim d_0}[Q_{\hat{i}}(s,\pi)]| \le 4V_{\max} \max_{i \in [m]\setminus\{i^\star\}} \frac{1}{\sigma_{\min}(A_{i,i^\star})} \varepsilon_{\mathsf{stat}}$$

$$= 24V_{\max}^3 \max_{i \in [m]\setminus\{i^\star\}} \frac{1}{\sigma_{\min}(A_{i,i^\star})} \sqrt{\frac{\log(4dm\delta^{-1})}{n}}.$$

$\square$

### C.3  Proof of Theorem 3

We bound

$$J(\pi) - \mathbb{E}_{d_0}\left[\widehat{Q}(s,\pi)\right] = \mathbb{E}_{d_0,\pi}\left[Q^\pi(s,a) - \widehat{Q}(s,a)\right]$$

$$= \frac{1}{1-\gamma} \mathbb{E}_{d^\pi}\left[Q^\pi(s,a) - \gamma Q^\pi(s',\pi) - \widehat{Q}(s,a) - \gamma\widehat{Q}(s',\pi)\right]$$

$$= \frac{1}{1-\gamma} \mathbb{E}_{d^\pi}\left[Q^\pi(s,a) - [\mathcal{T}^\pi Q^\pi](s,a) - \widehat{Q}(s,a) + \left[\mathcal{T}^\pi\widehat{Q}\right](s,a)\right]$$

$$= \frac{1}{1-\gamma} \mathbb{E}_{d^\pi}\left[\left[\mathcal{T}^\pi\widehat{Q}\right](s,a) - \widehat{Q}(s,a)\right]$$

$$\le \frac{1}{1-\gamma} \sqrt{\mathcal{C}^\pi \cdot \mathbb{E}_\mu\left[\left(\left[\mathcal{T}^\pi\widehat{Q}\right](s,a) - \widehat{Q}(s,a)\right)^2\right]}$$

where the second line follows from Bellman flow. Now we consider the term under the square root, and let $\widehat{g}_{\widehat{Q}} = \arg\min_{g \in \mathcal{G}_{\widehat{Q}}} \widehat{\ell}(g,\widehat{Q})$.

$$\mathbb{E}_\mu\left[\left(\left[\mathcal{T}^\pi\widehat{Q}\right](s,a) - \widehat{Q}(s,a)\right)^2\right] \le 2 \cdot \underbrace{\mathbb{E}_\mu\left[\left(\left[\mathcal{T}^\pi\widehat{Q}\right](s,a) - \widehat{g}_{\widehat{Q}}(s,a)\right)\right]^2}_{(T1)} + 2 \cdot \underbrace{\mathbb{E}_\mu\left[\left(\widehat{g}_{\widehat{Q}}(s,a) - \widehat{Q}(s,a)\right)^2\right]}_{(T2)}$$

We consider each term above individually. $(T1)$ is the regression error between $\widehat{g}_Q$ and the population regression solution $\mathcal{T}^\pi Q$, which we can control using well-established bounds. The second term $(T2)$ measure how close the Q-value is to its estimated Bellman backup. To bound these two terms we utilize the following results. The first controls the error between the squared-loss minimizer $\widehat{g}_Q$ and the population solution $\mathcal{T}^\pi Q$, and is adapted from [62].

**Lemma 1** (Lemma 9 from [62]). *Suppose that we have $|g|_\infty \le V_{\max}$ for all $g \in \mathcal{G}_Q$ and $Q \in \mathcal{Q}$, and define*

$$\widehat{g}_Q := \arg\min_{g \in \mathcal{G}_Q} \mathbb{E}_\mathcal{D}\left[(g(s,a) - r - \gamma Q(s',\pi))^2\right].$$

*Then with probability at least $1 - \delta$, for all $i \in [m]$ we have*

$$\mathbb{E}_\mu\left[(\widehat{g}_Q(s,a) - [\mathcal{T}^\pi Q](s,a))^2\right] \le \frac{16V_{\max}^2 \log\left(\frac{2m}{\delta}\right)}{n} := \varepsilon_{\mathsf{reg}}^2.$$

The second controls the error of estimating the objective for choosing $\widehat{i}$ from finite samples, and a proof is included at the end of this section.

**Lemma 2** (Objective estimation error). *Suppose that we have $\|g\|_\infty \le V_{\max}$ for all $g \in \mathcal{G}_Q$ and $Q \in \mathcal{Q}$. Then with probability at least $1 - \delta$, for all $g \in \mathcal{G}_Q$ and $Q \in \mathcal{Q}$ we have*

$$\max\left\{\frac{1}{2} \cdot \mathbb{E}_\mu\left[(g(s,a) - Q(s,a))^2\right] - \mathbb{E}_\mathcal{D}\left[(g(s,a) - Q(s,a))^2\right],\right.$$

$$\left.\mathbb{E}_\mathcal{D}\left[(g(s,a) - Q(s,a))^2\right] - \frac{3}{2} \cdot \mathbb{E}_\mu\left[(g(s,a) - Q(s,a))^2\right]\right\} \le \frac{3V_{\max}^2 \log\left(\frac{2m}{\delta}\right)}{n} := \varepsilon_{\mathsf{obj}}.$$

Using Lemma 1, we directly obtain that with probability at least $1 - \delta$,

$$(\text{T1}) \leq \varepsilon_{\text{reg}}^2.$$

By leveraging Lemma 2, we have that with probability at least $1 - \delta$,

$$
\begin{aligned}
(\text{T2}) &= \mathbb{E}_\mu\left[\left(\widehat{g}_{\widehat{Q}}(s,a) - \widehat{Q}(s,a)\right)^2\right] \\
&\leq 2 \cdot \varepsilon_{\text{obj}} + 2 \cdot \mathbb{E}_{\mathcal{D}}\left[\left(\widehat{g}_{\widehat{Q}}(s,a) - \widehat{Q}(s,a)\right)^2\right] \\
&\leq 2 \cdot \varepsilon_{\text{obj}} + 2 \cdot \mathbb{E}_{\mathcal{D}}\left[(\widehat{g}_{Q^\pi}(s,a) - Q^\pi(s,a))^2\right] \\
&\leq 4 \cdot \varepsilon_{\text{obj}} + 3 \cdot \mathbb{E}_\mu\left[(\widehat{g}_{Q^\pi}(s,a) - Q^\pi(s,a))^2\right] \\
&= 4 \cdot \varepsilon_{\text{obj}} + 3 \cdot \mathbb{E}_\mu\left[(\widehat{g}_{Q^\pi}(s,a) - [\mathcal{T}^\pi Q^\pi](s,a))^2\right] \\
&\leq 4 \cdot \varepsilon_{\text{obj}} + 3 \cdot \varepsilon_{\text{reg}}^2
\end{aligned}
$$

where in the first inequality we apply Lemma 2 (by lower bounding the LHS with the first expression in the $\max$); in the second we use the Q-value realizability assumption $Q^\pi \in \mathcal{Q}$ with the fact that $\widehat{Q}$ is the minimizer of the empirical objective; and in the third we again apply Lemma 2 (now lower bounding the LHS with the second expression in the $\max$). Then we use the identity that $Q^\pi = \mathcal{T}^\pi Q^\pi$, and apply the squared-loss regression guarantee. The bounds for (T1) and (T2) mean that

$$\mathbb{E}_\mu\left[\left(\left[\mathcal{T}^\pi \widehat{Q}\right](s,a) - \widehat{Q}(s,a)\right)^2\right] \leq 8\left(\varepsilon_{\text{obj}} + \varepsilon_{\text{reg}}^2\right),$$

resulting in the final estimation bound of

$$
\begin{aligned}
J(\pi) - \mathbb{E}_{d_0}\left[\widehat{Q}(s,\pi)\right] &\leq \frac{1}{1-\gamma}\sqrt{\mathcal{C}^\pi \cdot \mathbb{E}_\mu\left[\left(\left[\mathcal{T}^\pi \widehat{Q}\right](s,a) - \widehat{Q}(s,a)\right)^2\right]} \\
&\leq \frac{1}{1-\gamma}\sqrt{8 \cdot \mathcal{C}^\pi \cdot \left(\varepsilon_{\text{obj}} + \varepsilon_{\text{reg}}^2\right)}, \\
&= \frac{V_{\max}}{1-\gamma}\sqrt{\frac{152 \cdot \mathcal{C}^\pi \cdot \log\left(\frac{2m}{\delta}\right)}{n}},
\end{aligned}
$$

which holds with probability at least $1 - 2\delta$.

*Proof of Lemma 2.* Observe that the random variable $(g(s,a) - Q(s,a))^2 \in [-V_{\max}^2, V_{\max}^2]$, and

$$
\begin{aligned}
\mathbb{V}_\mu\left[(g(s,a) - Q(s,a))^2\right] &\leq \mathbb{E}_\mu\left[(g(s,a) - Q(s,a))^4\right] \\
&\leq V_{\max}^2 \cdot \mathbb{E}_\mu\left[(g(s,a) - Q(s,a))^2\right].
\end{aligned}
$$

Then, applying Bernstein's inequality with union bound, we have that, for any $g \in \mathcal{G}_Q$ and $Q \in \mathcal{Q}$ with probability at least $1 - \delta$,

$$
\begin{aligned}
&\left|\mathbb{E}_\mu\left[(g(s,a) - Q(s,a))^2\right] - \mathbb{E}_{\mathcal{D}}\left[(g(s,a) - Q(s,a))^2\right]\right| \\
&\leq \sqrt{\frac{4\mathbb{V}_\mu\left[(g(s,a) - Q(s,a))^2\right]\log\left(\frac{2m}{\delta}\right)}{n}} + \frac{V_{\max}^2 \log\left(\frac{2m}{\delta}\right)}{n} \\
&\leq \sqrt{\frac{4V_{\max}^2\mathbb{E}_\mu\left[(g(s,a) - Q(s,a))^2\right]\log\left(\frac{2m}{\delta}\right)}{n}} + \frac{V_{\max}^2 \log\left(\frac{2m}{\delta}\right)}{n} \\
&\leq \frac{\mathbb{E}_\mu\left[(g(s,a) - Q(s,a))^2\right]}{2} + \frac{3V_{\max}^2 \log\left(\frac{2m}{\delta}\right)}{n}.
\end{aligned}
$$

Expanding the absolute value on the LHS and rearranging, this then implies that

$$\frac{1}{2} \cdot \mathbb{E}_\mu\Big[(g(s,a) - Q(s,a))^2\Big] \leq \mathbb{E}_\mathcal{D}\Big[(g(s,a) - Q(s,a))^2\Big] + \frac{3V_{\max}^2 \log\big(\frac{2m}{\delta}\big)}{n},$$

$$\mathbb{E}_\mathcal{D}\Big[(g(s,a) - Q(s,a))^2\Big] \leq \frac{3}{2} \cdot \mathbb{E}_\mu\Big[(g(s,a) - Q(s,a))^2\Big] + \frac{3V_{\max}^2 \log\big(\frac{2m}{\delta}\big)}{n}.$$

Combining these statements completes the proof. □

### C.4 Proof of Theorem 4

We bound

$$
\begin{aligned}
J(\pi) - \mathbb{E}_{d_0}\Big[\widehat{Q}(s,\pi)\Big] &= \mathbb{E}_{d_0,\pi}\Big[Q^\pi(s,a) - \widehat{Q}(s,a)\Big] \\
&= \frac{1}{1-\gamma} \mathbb{E}_{d^\pi}\Big[Q^\pi(s,a) - \gamma Q^\pi(s',\pi) - \widehat{Q}(s,a) - \gamma\widehat{Q}(s',\pi)\Big] \\
&= \frac{1}{1-\gamma} \mathbb{E}_{d^\pi}\Big[Q^\pi(s,a) - [\mathcal{T}^\pi Q^\pi](s,a) - \widehat{Q}(s,a) + \big[\mathcal{T}^\pi\widehat{Q}\big](s,a)\Big] \\
&= \frac{1}{1-\gamma} \mathbb{E}_{d^\pi}\Big[\big[\mathcal{T}^\pi\widehat{Q}\big](s,a) - \widehat{Q}(s,a)\Big] \\
&\leq \frac{\mathcal{C}_\infty^\pi}{1-\gamma} \cdot \mathbb{E}_\mu\Big[\big|\big[\mathcal{T}^\pi\widehat{Q}\big](s,a) - \widehat{Q}(s,a)\big|\Big] \\
&\leq \max_{g \in \mathcal{G}_{\widehat{Q}}} \mathbb{E}_\mu\Big[\mathrm{sgn}\big(\widehat{Q}(s,a) - g(s,a)\big)\big(\widehat{Q}(s,a) - r - \gamma\widehat{Q}(s',\pi)\big)\Big]
\end{aligned}
$$

By assumption, $\max_{q \in \mathcal{Q}} \|q\|_\infty \leq V_{\max}$, and similarly $\max_{g \in \mathcal{G}_Q} \|g\|_\infty \leq V_{\max}$ for all $Q \in \mathcal{Q}$. Then for any $Q \in \mathcal{Q}$ and $g \in \mathcal{G}_Q$ and $(s,a) \in \mathcal{S} \times \mathcal{A}$ and $r \in [0, R_{\max}]$,

$$\mathrm{sgn}(Q(s,a) - g(s,a))(Q(s,a) - r - \gamma Q(s',\pi)) \in [-V_{\max}, V_{\max}],$$

and, using Hoeffding's inequality, we have for all $Q \in \mathcal{Q}$ and $g \in \mathcal{G}_Q$ that, with probability at least $1 - \delta$,

$$
\begin{aligned}
&\Big|\mathbb{E}_\mu\Big[\mathrm{sgn}\big(\widehat{Q}(s,a) - g(s,a)\big)\big(\widehat{Q}(s,a) - r - \gamma\widehat{Q}(s',\pi)\big)\Big] \\
&- \mathbb{E}_\mathcal{D}\Big[\mathrm{sgn}\big(\widehat{Q}(s,a) - g(s,a)\big)\big(\widehat{Q}(s,a) - r - \gamma\widehat{Q}(s',\pi)\big)\Big]\Big| \leq 2V_{\max}\sqrt{\frac{\log\big(\frac{2m}{\delta}\big)}{n}} := \varepsilon_{\mathsf{obj}}.
\end{aligned}
$$

Then using this concentration in the last line of the previous block,

$$
\begin{aligned}
J(\pi) - \mathbb{E}_{d_0}\Big[\widehat{Q}(s,\pi)\Big] &\leq \max_{g \in \mathcal{G}_{\widehat{Q}}} \mathbb{E}_\mathcal{D}\Big[\mathrm{sgn}\big(\widehat{Q}(s,a) - g(s,a)\big)\big(\widehat{Q}(s,a) - r - \gamma\widehat{Q}(s',\pi)\big)\Big] + \varepsilon_{\mathsf{obj}} \\
&\leq \max_{g \in \mathcal{G}_{Q^\pi}} \mathbb{E}_\mathcal{D}[\mathrm{sgn}(Q^\pi(s,a) - g(s,a))(Q^\pi(s,a) - r - \gamma Q^\pi(s',\pi))] + \varepsilon_{\mathsf{obj}} \\
&\leq \max_{g \in \mathcal{G}_{Q^\pi}} \mathbb{E}_\mu[\mathrm{sgn}(Q^\pi(s,a) - g(s,a))(Q^\pi(s,a) - r - \gamma Q^\pi(s',\pi))] + 2 \cdot \varepsilon_{\mathsf{obj}} \\
&= \max_{g \in \mathcal{G}_{Q^\pi}} \mathbb{E}_\mu[\mathrm{sgn}(Q^\pi(s,a) - g(s,a))(Q^\pi(s,a) - [\mathcal{T}^\pi Q^\pi](s,a))] + 2 \cdot \varepsilon_{\mathsf{obj}} \\
&= \max_{g \in \mathcal{G}_{Q^\pi}} \mathbb{E}_\mu[\mathrm{sgn}(Q^\pi(s,a) - g(s,a))(Q^\pi(s,a) - Q^\pi(s,a))] + 2 \cdot \varepsilon_{\mathsf{obj}} \\
&= 2 \cdot \varepsilon_{\mathsf{obj}},
\end{aligned}
$$

where in the first and third inequalities we apply the above concentration inequality, and in the second inequality we use the fact that $\widehat{Q}$ is the minimizer of the empirical objective, i.e.,

$$\widehat{Q} = \arg\min_{Q \in \mathcal{Q}} \max_{g \in \mathcal{G}_Q} \mathbb{E}_\mathcal{D}[\mathrm{sgn}(Q(s,a) - g(s,a))(Q(s,a) - r - Q(s',\pi))].$$

Combining the above inequalities, we obtain the theorem statement,

$$J(\pi) - \mathbb{E}_{d_0}\Big[\widehat{Q}(s,\pi)\Big] \leq 2 \cdot \mathcal{C}_\infty^\pi \cdot \varepsilon_{\mathsf{obj}}.$$

### C.5 Further Discussion on Comparison to BVFT and the Coverage Assumptions

Besides the difference in rates, the guarantees for LSTD-Tournament and BVFT differ in their coverage assumptions, which also differ from those for the model-based selectors (Theorems 3 and 4). In fact, Theorems 3 and 4 use the standard "concentrability coefficient" $\mathcal{C}_\infty^\pi$ that is widely adopted in offline RL theory [6, 10]. In contrast, LSTD-Tournament's coverage parameter is $\max_{i \in [m] \setminus \{i^\star\}} \frac{1}{\sigma_{\min}(A_{i,i^\star})}$, which is a highly specialized notion of coverage inherited from that of LSTDQ ($\sigma_{\min}(A)$ from Theorem 1). Similarly, the coverage parameter of BVFT is also a notion of coverage (called "aggregated concentrability" [62, 19]) specialized to state abstractions. As far as we know there is no definitive relationship between the coverage parameters of LSTD-Tournament and BVFT and neither dominates the other. Furthermore, these non-standard, highly algorithm-specific coverage definitions are a common situation for algorithms that only require realizability (instead of Bellman-completeness) as their expressivity condition, as discussed in Jiang and Xie [23].

**Problems in $\sigma_{\min}(A)$ as LSTDQ's Coverage Assumption.**   As mentioned above, the coverage condition of LSTD-Tournament is inherited from its "base" algorithm, i.e., the $\sigma_{\min}(A)$ term in LSTDQ. Despite its wide use in the analyses of LSTDQ algorithms [30, 3, 48], we discover several problems with this definition that warrants further investigation. First, the quantity is not invariant to reparameterization, in the sense that if we scale the linear features $\phi$ with a constant (and scale $\theta$ accordingly), $\sigma_{\min}(A)$ will also change despite that the estimation problem essentially remains the same. Second, if $\phi$ has linearly dependent coordinates, $\sigma_{\min}(A)$ can be 0 while this says nothing about the quality of data (which could have perfect coverage); this is usually not a concern in LSTDQ analysis since we can always pre-process the features, but for its application in LSTD-Tournament the lack of linear independence can happen if we have nearly identical candidate functions $Q_i \approx Q_j$. Last but not least, we lack good understanding of when $\sigma_{\min}(A)$ is well-behaved outside very special cases; for example, it is well-known that fully on-policy data (in the sense that $\mu$ is invariant to the transition dynamics under $\pi$) implies lower-bounded $\sigma_{\min}(A)$. However, as soon as $\mu$ is off-policy, there lacks general characterization of what kind of $\mu$ helps lower-bound $\sigma_{\min}(A)$. For example, when concentrability coefficient $\mathcal{C}_\infty^\pi$ is well bounded, it is unclear whether that implies lower-bounded $\sigma_{\min}(A)$.

In this project we have explored ways to mitigate these issues, such as defining coverage as $\sigma_{\min}(\Sigma)/\sigma_{\min}(A)$ to avoid the linear-dependence issue. However, none of the resolutions we tried solve all problems elegantly. This question is, in fact, quite orthogonal to the central message of our work and of separate interest in the fundamental theory of RL, and speaks to how algorithms like LSTDQ are seriously under-investigated. In fact, coverage in state abstractions has been similarly under-investigated and has only seen interesting progress recently [19]. We wish to investigate the question for LSTDQ in the future, and any progress there can be directly inherited to improve the analysis of LSTD-Tournament.

## D   Experiment Details

### D.1   Environment Setup: Noise and State Resetting

**State Resetting.**   Monte-Carlo rollouts for Q-value estimation rely on the ability to (re)set the simulator to a particular state from the offline dataset. To the best of our knowledge, Mujoco environment does not natively support state resetting, and assigning values to the observation vector does not really change the underlying state. However, state resetting can still be implemented by manually assigning the values of the position vector `qpos` and the velocity vector `qvel`.

**Noise.**   As mentioned in Section 6, we add noise to Hopper to create more challenging stochastic environments and create model selection tasks where candidate simulators have different levels of stochasticity. Here we provide the details about how we inject randomness into the deterministic dynamics of Hopper. Mujoco engine realizes one-step transition by leveraging `mjData.{ctrl, qfrc_applied, xfrc_applied}` objects [13], where `mjData.ctrl` corresponds to the action taken by our agent, and `mjData.{qfrc_applied, xfrc_applied}` are the user-defined perturbations in the joint space and Cartesian coordinates, respectively. To inject randomness into the transition at a noise level of $\sigma$, we first sample an isotropic Gaussian noise with variance $\sigma^2$ as the stochastic force in `mjData.xfrc[:3]` upon each transition, which jointly determines the next state with the input action `mjData.ctrl`, leaving the joint data `mjData.qfrc` intact.

## D.2 Experiment Settings

**MF/MB.G/N.** The settings of different experiments are summarized in Table 1. We first run DDPG in the environment of $g = -30, \sigma = 32$, and obtain 15 deterministic policies $\{\pi_{0:14}\}$ from the checkpoints. The first 10 are used as target policies in MF.G/N experiments, and MB.G/N use fewer due to the high computational cost. For the main results (Section 6.1), the choice of $M^\star$ is usually the two ends plus the middle point of the grid ($\mathcal{M}_\mathbf{g}$ or $\mathcal{M}_\mathbf{n}$). The corresponding behavior policy is an epsilon-greedy version of one of the target policies, denoted as $\pi_i^\epsilon$, which takes the deterministic action of $\pi_i(s)$ with probability 0.7, and add a unit-variance Gaussian noise to $\pi_i(s)$ with the remaining 0.3 probability.

**MF/MB.Off.G/N.** In the above setup, the behavior and the target policies all stem from the same DDPG training procedure. While these policies still have significant differences (see Figure 2L), the distribution shift is relatively mild. For the data coverage experiments (Section 6.3), we prepare a different set of behavior policies that intentionally offer poor coverage: these policies, denoted as $\pi_i^{\mathrm{poor}}$, are obtained by running DDPG with a different neural architecture (than the one used for generating $\pi_{0:14}$) in a different environment of $g = -60, \sigma = 100$. We also provide the parallel of our main experiments in Figures 3 and 4 under these behavior policies with poor coverage in Appendix E.1.

**MF.T.G.** This experiment is for data coverage (Section 6.3), where $\mathcal{D}$ is a mixture of two datasets, one sampled from $\pi_7$ (which is the sole target policy being considered) and one from $\pi_i^{\mathrm{poor}}$ that has poor coverage. They are mixed together under different ratios as explained in Section 6.3.

| | **Gravity g** | **Noise Level $\sigma$** | **Groundtruth Model $M^\star$ and Behavior Policy $\pi_b$** | **Target Policies $\Pi$** |
|---|---|---|---|---|
| MF.G | LIN$(-51, -9, 15)$ | 100 | $\{(M_i, \pi_i^\epsilon), i \in \{0, 7, 14\}\}$ | $\{\pi_{0:9}\}$ |
| MF.N | -30 | LIN$(10, 100, 15)$ | $\{(M_i, \pi_i^\epsilon), i \in \{0, 7, 14\}\}$ | $\{\pi_{0:9}\}$ |
| MB.G | LIN$(-36, -24, 5)$ | 100 | $\{(M_i, \pi_i^\epsilon), i \in \{0, 2, 4\}\}$ | $\{\pi_{0:5}\}$ |
| MB.N | -30 | LIN$(10, 100, 5)$ | $\{(M_i, \pi_i^\epsilon), i \in \{0, 2, 4\}\}$ | $\{\pi_{0:5}\}$ |
| MF.OFF.G | LIN$(-51, -9, 15)$ | 100 | $\{(M_i, \pi_i^{\mathrm{poor}}), i \in \{0, 7, 14\}\}$ | $\{\pi_{0:9}\}$ |
| MF.OFF.N | -30 | LIN$(10, 100, 15)$ | $\{(M_i, \pi_i^{\mathrm{poor}}), i \in \{0, 7, 14\}\}$ | $\{\pi_{0:9}\}$ |
| MF.T.G | LIN$(-51, -9, 15)$ | 100 | $\{(M_i, \pi_8 \,\&\, \pi_i^{\mathrm{poor}}), i \in \{0, 7, 14\}\}$ | $\{\pi_8\}$ |

Table 1: Details of experiment settings. LIN$(a, b, n)$ (per numpy convention) refers to the arithmetic sequence with $n$ elements, starting from $a$ and ending in $b$ (e.g. LIN$(0, 1, 6) = \{0, 0.2, 0.4, 0.6, 0.8, 1.0\}$).

## D.3 Computational Resources

We here provide a brief discussion of the amount of computation used to produce the results. The major cost is in Q-caching, i.e., rolling out trajectories. The cost consists of two parts: environment simulation steps (CPU-intensive) and neural-net inference for policy calls (GPU-intensive). The main experiment on **MF.G**, took nearly a week on a 4090 PC. The runtimes of **MF.N**, **MB.G**, **MB.N** are comparable to **MF.G**.

# E  Additional Experiment Results

**Subgrid Studies.** Figures 7 and 6 show more complete results for investigating the sensitivity to misspecification and gaps in Section 6.2 across 4 settings (good/poor coverage and gravity/noise grid).

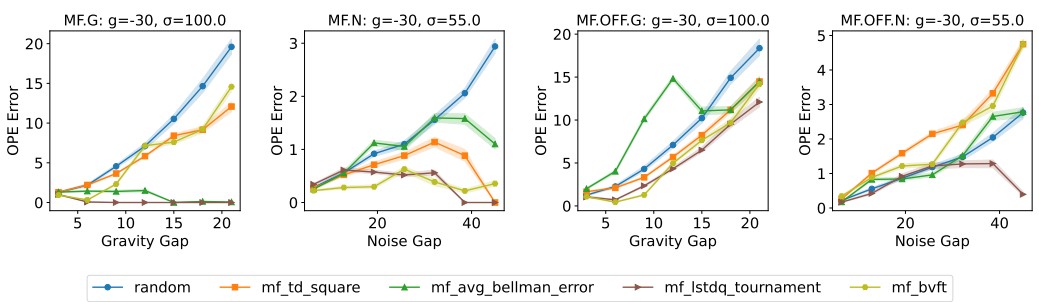

Figure 6: Subgrid studies for gaps. Plot **MF.N** is identical to Figure 5L.

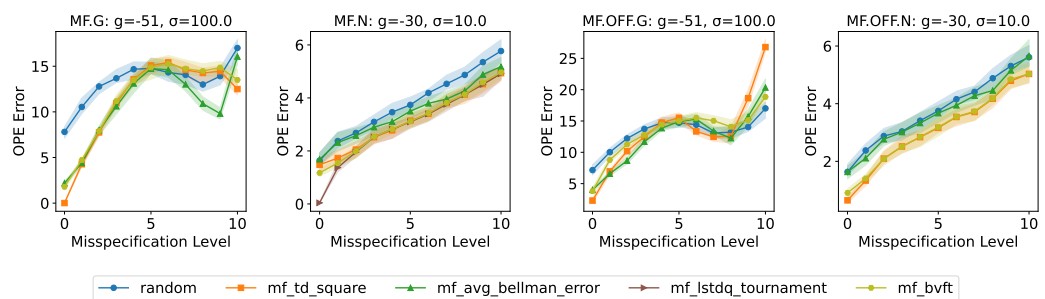

Figure 7: Subgrid studies for misspecification. Plot **MF.N** is identical to Figure 5M.

**Data Coverage.** Figure 8 shows more complete results for the data coverage experiment in Section 6.3, including more choices of $M^\star$.

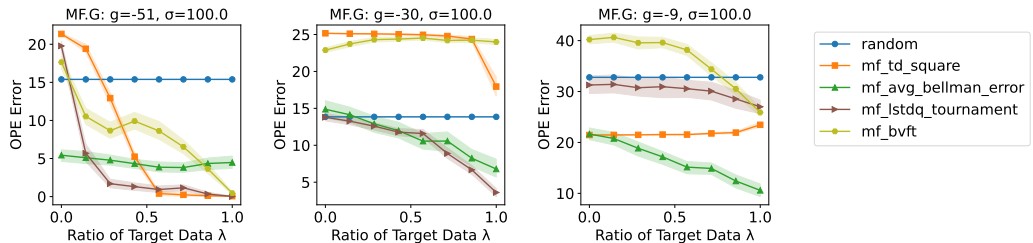

Figure 8: Data coverage results. The left figure is identical to Figure 5L.

### E.1 Poor Coverage Results

We now show the counterpart of our model-free main results (Figure 3) under behavior policies that offer poor coverage. This makes the problem very challenging and no single algorithm have strong performance across the board. For example, naïve model-based demonstrate strong performance in **MF.OFF.G** (top row of Figure 9) and resilience to poor coverage, while still suffers catastrophic failures in **MF.OFF.N**. While LSTD-Tournament generally is more reliable than other methods, it also has worse-than-random performance in one of the environments in **MF.OFF.G**.

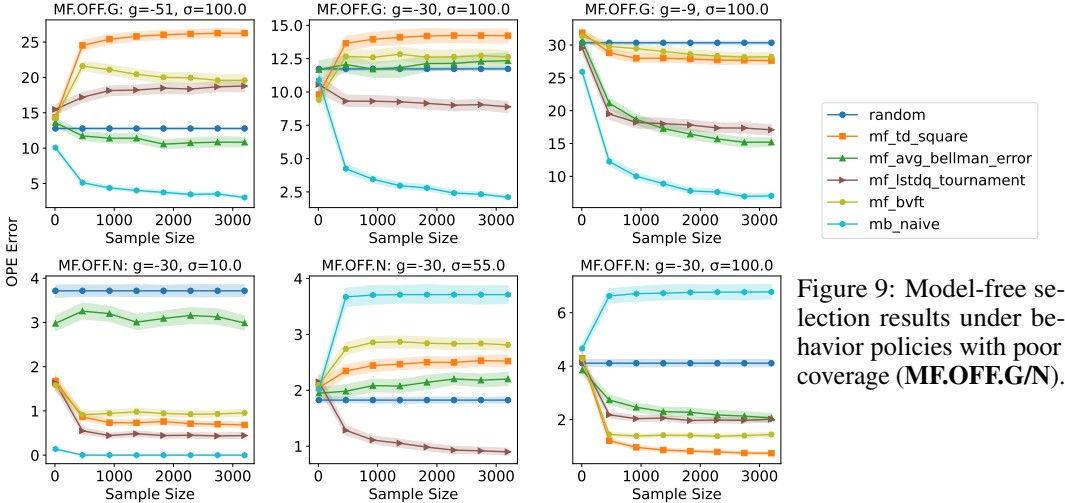

Figure 9: Model-free selection results under behavior policies with poor coverage (**MF.OFF.G/N**).

## E.2 LSTDQ Family

As mentioned at the end of Section 3, our LSTD-Tournament can have several variants depending on how we design and transform the linear features. Here we compare 3 of them in Figure 10. The LSTD-Tournament method in all other figures corresponds to the "normalized_diff" version.

- **Vanilla:** $\phi_{i,j} = [Q_i, Q_j]$.
- **Normalized:** $\phi_{i,j} = [Q_i/c_i, Q_j/c_j]$, where $c_i = \sqrt{\mathbb{V}_{(s,a)\sim\mu}[Q_i(s,a)]}$ normalizes the discriminators to unit variance on the data distribution. In practice these variance parameters are estimated from data.
- **Normalized_diff:** $\phi_{i,j} = [Q_i/c_i, (Q_j - Q_i)/c_{j,i}]$, where $c_i$ and $c_{j,i}$ performs normalization in the same way as above.

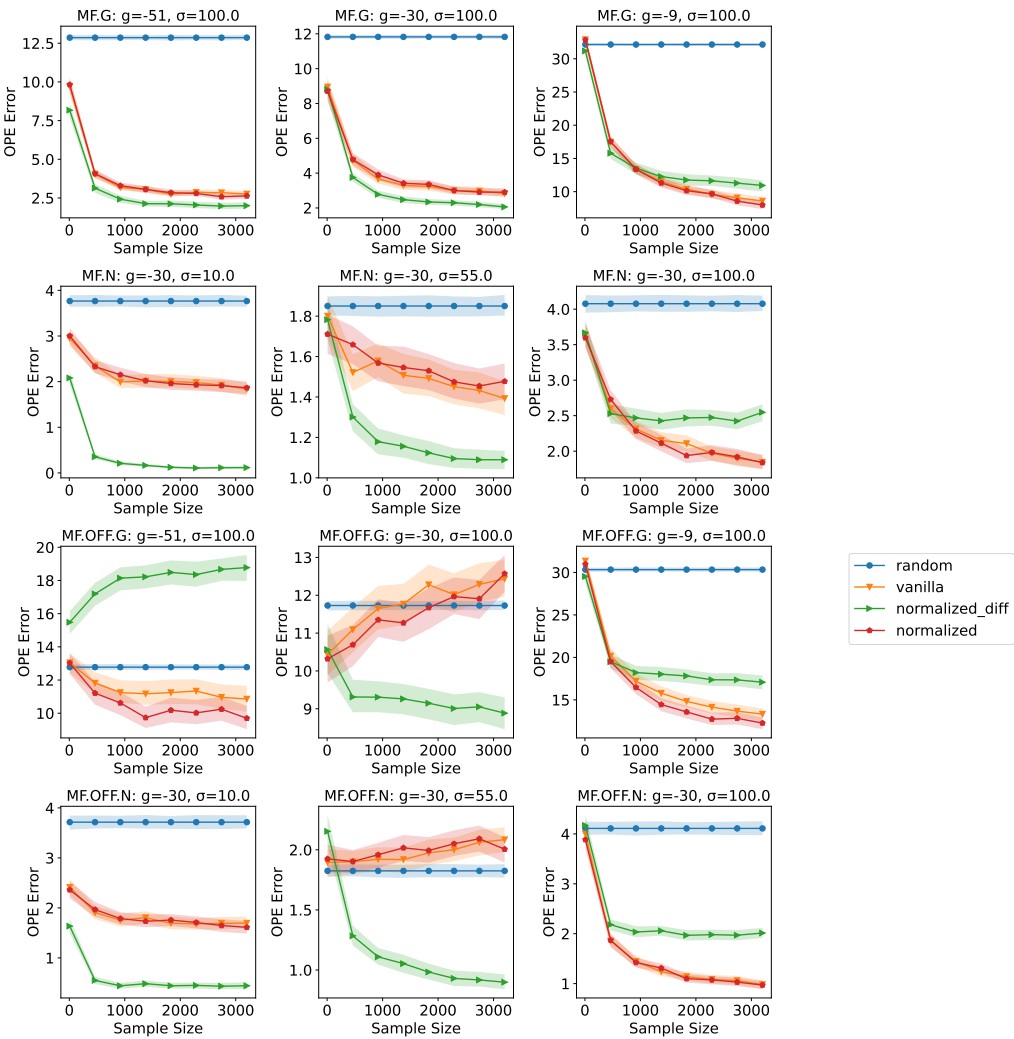

Figure 10: Comparison of variants of LSTD-Tournament.

### E.3 $O(1)$ **Rollouts**

In our experiment design, we use a fairly significant number of rollouts $l = 128$ to ensure relatively accurate estimation of the Q-values. However, for the average Bellman error and the LSTD-Tournament algorithms, they enjoy convergence even when $l$ is a constant. For example, consider the average Bellman error:

$$\mathbb{E}_{\mathcal{D}}[Q_i(s, a) - r - \gamma Q_i(s', \pi)],$$

which is an estimation of $\mathbb{E}_{\mu}[Q_i(s, a) - r - \gamma Q_i(s', \pi)]$. Thanks to its linearity in $Q_i$, replacing $Q_i$ with its few-rollout (or even single-rollout) Monte-Carlo estimates will leave the unbiasedness of the estimator intact, and Hoeffding's inequality implies convergence as the sample size $n = |\mathcal{D}|$ increases, even when $l$ stays as a constant, which is an advantage compared to other methods. That said, in practice, having a relatively large $l$ can still be useful as it reduces the variance of each individual random variable that we average across $\mathcal{D}$, and the effect can be significant when $n$ is relatively small.

A similar but slightly more subtle version of this property also holds for LSTD-Tournament. Take the vanilla version in Section 3 as example, we need to estimate

$$\mathbb{E}_{\mathcal{D}}[Q_j(s, a)(Q_i(s, a) - r - \gamma Q_i(s', \pi))].$$

Again, we can replace $Q_j$ and $Q_i$ with their Monte-Carlo estimates, as long as the Monte-Carlo trajectories for $Q_i$ and $Q_j$ are independent. This naturally holds in our implementation when $j \neq i$, but is violated when $j = i$ since $Q_j(s, a)$ and $Q_i(s, a)$ will share the same set of random rollouts, leading to biases. A straightforward resolution is to divide the Monte-Carlo rollouts into two sets, and $Q_j(s, a)$ and $Q_i(s, a)$ can use different sets when $j \neq i$. We empirically test this procedure in Figure 11, where the OPE errors of average Bellman error and different variants of LSTD-Tournament are plotted against the number of rollouts in each set (i.e., $l/2$). While a small number of rollouts can sometimes lead to reasonable performance, more rollouts are often useful in providing further variance reduction and hence more accurate estimations.

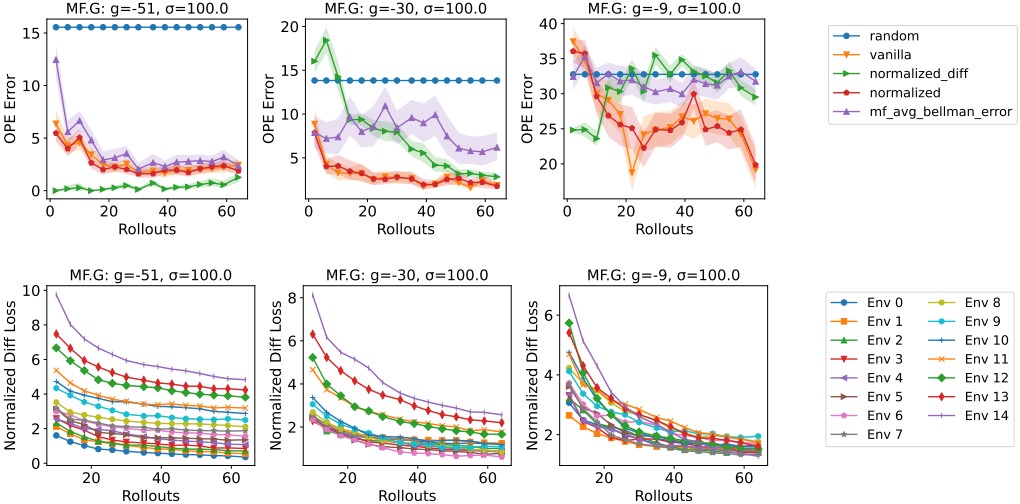

Figure 11: The effect of small rollouts in LSTD-Tournament methods. Sample size is fixed at $n = 3200$ and only $l$ (the number of rollouts) varies. Since the rollouts are divided into two separate sets to ensure independence, each Q-value is estimated using $l/2$ rollouts in this experiment, which is shown on the x-axes. The top row shows the OPE error (i.e., final performance), whereas the bottom row shows the convergence of loss estimates.

