# OpenReview forum: "Model Selection for Off-policy Evaluation: New Algorithms and Experimental Protocol"
_NeurIPS.cc/2025/Conference — NeurIPS 2025 poster_

### Official Review · Reviewer_ZupL · 2025-06-27

**Clarity:** 3
**Significance:** 3
**Originality:** 3
**Rating:** 4
**Confidence:** 4

**Summary:**

This paper addresses a fundamental challenge in offline RL that inhibits its practical applicability: How does one select the best off-policy evlauation (OPE) method? If a trusted simulator of the environment is not available, it is challenging to select a candidate of methods to evaluate a set of trained policies. The authors contribute bits to a multitude of topics including the introduction of selection algorithms (LSTD-Tournament, model-based selectors), theoretical analysis, an experiment protocol and empirical validation on the Gym-Hopper environment.

**Questions:**

(1) Can you restructure the paper in a way that Figure 5 is included in the paper instead of the appendix? To me, Figure 5 is a vital part to understand the paper. (could help B, C, D)

(2) Is "model selection for OPE" an established term? If so, can you provide references to this, and if not, can you elaborate why the term is appropriate, especially with the multitude of meanings of the phrase "model" in the OPE context? (B, D)

(3) Can you elaborate more in-depth your observations of model-based selectors? It is surprising to me that they do not hold your expectations. (F)

(4) Can you improve the clarity in the abstract and in the introduction (especially the first paragraph) to better reflect the topics of the paper? (B, D)

(5) Can you include a small conclusion, maybe would fit the start of section 7, that wraps up the paper? (D, E)



I will reconsider my "Rating" if these questions are addressed.

**Ethical Concerns:**

["NO or VERY MINOR ethics concerns only"]

**Final Justification:**

Considering the rebuttal and the various discussions between authors and the other reviewers, I increased my score to **4: Borderline accept**.

The paper covers ideas in a promising direction, highlighting a gap in the current literature that is under-researched despite its important practical implications. While I'm inclined to be more positive about the paper, I totally agree with the critique that the paper's presentation leaves room for improvement. With the overall organisation, selection of topics and a rather vague central thread in mind, the paper seems difficult to digest.

Overall, I'm still in favour of this submission's acceptance.

**Limitations:**

yes

**Paper Formatting Concerns:**

No major formatting issues observed.

**Quality:**

3

**Strengths And Weaknesses:**

Major Weaknesses:

(A) The combination/selections of contributions is to my understanding rather unconventional. (Quality-related)

(B) The phrasing of "what the paper is about" lacks clarity, and in my opinion is prone to be misunderstood. (Clarity-related)

(C) An essential part, namely Figure 5, that is vital to the comprehensibility of the paper is pushed into the appendix. (Quality/Clarity-related)

Minor Weaknesses:

(D) Due to the high-level perspective, the topic is generally hard to grasp.

(E) The paper lacks a concise conclusion which wraps up the multitude of topics. It has a thorough exemplification of its protocol in 6 and limitations/future work in 7.

(F) The discussion of the results in 6 could be clearer.

Strengths:

(G) The paper addresses an important and understudied gap in the literature.

(H) The paper presents significant insights, generally the empirical resutls of model-based and model-free methods is very appreciated and especially the presented subgrid studies are interesting to me.


Please address the points under "Questions".

---

> ### Author Rebuttal · Authors · 2025-07-27
>
> We thank the reviewer for the helpful comments and feedback. We note that you rated “3: good” in all subcategories but gave “borderline reject” in overall rating. We hope you can reconsider the rating after seeing our response and are glad that you are open to it (“I will reconsider…”).
>
> ---
>
> **(A) The combination/selections of contributions is to my understanding rather unconventional. (Quality-related)**
>
> We agree that the combination of contributions is unconventional (theory + experiment protocol + empirical), but respectfully disagree that this hurts the quality of the paper. These contributions emerge naturally when we investigate the topic: after developing the methods and their theories, we want to empirically test them, but the protocols in the prior works have been hard to use and caused many troubles in our replication (see e.g., Footnote 1 on page 2). So we come up with a new protocol that is much better in producing stable candidates and enable interesting studies (Sections 6.2 and 6.3) that aren’t possible before.
>
> ---
>
> **(1) Restructure the paper to include Figure 5 in main text**
>
> Thanks for the great suggestion. We totally agree that Figure 5 can be very helpful in understanding the overall problem and protocols, especially given that the complications in offline evaluation and selection are not something that most of the RL researchers have thought about carefully. We had to defer the figure to the appendix due to space limit, but if the paper is accepted, we fully intend to include it in the main text given the 1 extra content page.
>
> ---
>
> **(2) Is “Model selection in OPE” an established term? especially with the multitude of meanings of the phrase "model"**
>
> We agree that the word “model” is heavily overloaded in OPE and RL in general. As we mentioned, model selection in OPE is a heavily under-investigated problem, so it’s hard to say it’s “well-established” given the scarcity of research in this area in the first place, but prior work like [62] did also use “offline model selection”. In addition, despite the multitude of meanings of “model”, “model selection” as a whole is a well-established technical term in statistics, and we are following this convention. In machine learning, model selection is closely related to concepts like cross validation and hyperparameter tuning (which we mention in the first sentence of abstract to help resolve the ambiguity of “model selection”), but the latter two refer to something more specific (e.g., cross validation refers to procedures like k-fold, and hyperparameter tuning often applies to settings where candidate functions/predictors are generated by the same algorithm with different hyperparameters) that do not necessarily apply to our problem setting, and the term “model selection” is the more general and abstract term which we think fit better.
>
> ---
>
> **(3) surprising to me that [model-based predictors] do not hold your expectations.**
>
> Indeed! Our rationale was that model-based predictors enjoy strong theoretical motivations: they apply to narrower settings (can only select from models, not Q-functions; Line 105), but leverage more information from the model (the Bellman operators); in particular, not having access to Bellman operators is _the_ biggest difficulty in model-free selection (Line 207), but the operators can be learned quite straightforwardly in the model-based setting (Line 208). **As a general wisdom, methods that properly leverage additional side information often perform better.** Moreover, model-based selectors enjoy more standard and interpretable guarantees found in the offline RL theory literature (Line 360). So our anticipation was that model-based predictors would outperform model-free ones, which is a trade-off between performance (model-based wins) and applicability (model-free wins). Therefore, the fact that model-based performs worse is definitely surprising, and **we view this a great opportunity to report interesting _negative_ results, which we hope is more appreciated and encouraged by the community**, and the surprisingly poor performance of model-based selectors further highlights the effectiveness of LSTD-tournament.
>
> As for why model-based selectors fail, we have had some preliminary investigation into this, which we can add to the appendix; one possible reason is that these estimators are pretty sensitive to small errors (e.g., the Monte-Carlo estimates still has small nontrivial errors despite we use $l=128$ rollouts (Line 288) which is quite computationally intensive (Appendix D.3)), and the model-based estimator suffers more due to its multi-stage process. In contrast, LSTD-tournament has an interesting property that it is largely immune to the Monte-Carlo errors (Appendix E.3), since the noise in each Q-estimate will be averaged out over the entire dataset due to the lack of absolute value or square inside the loss function (Eq.(10)).
>
> ---
>
> **(4) Can you improve the clarity in the abstract and in the introduction (especially the first paragraph) to better reflect the topics of the paper? (B, D)**
>
> We are definitely happy to but it is unclear which part causes confusion from your review. In particular, your summary of the paper seems spot on and we don’t see any obvious misunderstanding in your text. If you could elaborate more on where it needs clarification, we can improve the clarity of the text accordingly.
>
> ---
>
> **(5) Can you include a small conclusion, maybe would fit the start of section 7, that wraps up the paper? (D, E)**
>
> Definitely happy to and we plan to do so if the paper is accepted and given one extra content page.

---

> > ### Comment · Reviewer_ZupL · 2025-08-06
> >
> > Dear Authors,
> > Thank you for your detailed rebuttal and for addressing the concerns I previously raised. I would like to provide further clarity on some points.
> >
> > (A) I acknowledge your position that the combination of theory, experiment protocol, and empirical contributions is unconventional yet natural in the course of your research. However, I maintain that in this specific case, the unconventional nature does impact the quality of the paper. While such emergence of contributions is a part of researching unexplored areas, it is not a sufficient argument to justify their combination in a publication.
> >
> > (4) Regarding the clarity and structure:
> > Abstract and Introduction: The phrasing in the abstract, particularly beginning with "Holdout validation and hyperparameter tuning" initially generated expectations misaligned with the paper's content. The proximity of terms such as "model selection" and "model-based" methods led to initial confusion. Maybe a rephrasing approach where the paper's scope is outlined in a narrow-to-broad manner (instead of broad-to-narrow) could help the Abstract. The combination of (A) and (3) certainly impacts this point as well.
> >
> > Thank you for addressing points (1) and (5) regarding the restructuring and inclusion of Figure 5 and the conclusion. These changes are crucial and a necessity for me to increase my score.
> >
> > Currently, I intend to increase my score from 3 to 4, meaning I lean to accept the paper.

---

> > > ### Author Response · Authors · 2025-08-06
> > >
> > > Dear reviewer ZupL,
> > >
> > > Thanks for your acknowledgement of our rebuttal and the further comments! We understand your remaining concerns and will also adjust the abstract and introduction, especially the use of wording, accordingly.
> > >
> > > thanks,
> > >
> > > Authors

---

> > > ### Comment · Area_Chair_n77f · 2025-08-08
> > >
> > > Dear reviewer,
> > >
> > > Thanks for engaging with the authors. Can you please also use the acknowledge-response button to confirm that you've seen the discussion phase instructions?
> > >
> > > Thanks,
> > > Your AC

---

> ### Author Response · Authors · 2025-08-05
>
> Dear reviewer ZupL,
>
> This is gentle reminder that the time for author-reviewer discussion is running out. We'd very much appreciate your consideration of rebuttal and further comments (e.g., which aspects of the abstracts that you want to see more clarifications) so that we have an opportunity to better address your concerns.
>
> thanks,
>
> Authors

---

### Official Review · Reviewer_2FCh · 2025-07-02

**Clarity:** 3
**Significance:** 3
**Originality:** 3
**Rating:** 5
**Confidence:** 2

**Summary:**

The paper studies the model selection problem in offline reinforcement learning. The paper studies both model-free and model-based selectors for off-policy evaluation (OPE) with theoretical guarantees and simulated experiments. For model-free selectors, the paper proposes a novel LSTD-Tournament, which exbits a smaller error rate compared to existing method, i.e., BVFT. For model-based methods, the authors studied regression-based selectors and sign-flip selectors.

**Questions:**

1. Maybe I'm unfamiliar with relevant literature, but can you obtain lower bounds for model-free and model-based error rates? How tight are your proposed methods?

2. In figure 2, can you explain why mb_naive performs well in the top row but has disastorous error in the last two in bottom row?

**Ethical Concerns:**

["NO or VERY MINOR ethics concerns only"]

**Final Justification:**

See the discussion with authors.

**Limitations:**

The authors addressed the limitations.

**Quality:**

3

**Strengths And Weaknesses:**

Strenths:
1. The paper studies an interesting model selection problem for OPE instances. Both types of methods have nice theoretical guarantees compared to baselines.

2. The experiments are also extensive and support the theoretical claims.

Weaknesses:

1. As the authors also stated, the assumption that Q-function candidates may be unrealistic in practice.

2. In Section 5, even though the theorems are deferred to appendix, I suggest the authors to summarize them briefly in the main text.

---

> ### Author Rebuttal · Authors · 2025-07-27
>
> We thank the reviewer for the positive evaluation, and respond to your questions below.
>
> ---
>
> **Maybe I'm unfamiliar with relevant literature, but can you obtain lower bounds for model-free and model-based error rates? How tight are your proposed methods?**
>
> If by rates you mean the dependence on n, then $n^{-1/2}$ is the best you can get for simple reasons: OPE is clearly more general than estimating the mean of a bounded random variable and subsumes the latter as a special case. For mean estimation, without additional structures and assumptions, the $n^{-1/2}$ rate is the minimax rate and well-established in the statistics literature, so it is simply impossible to do better than that in general. That said, our bounds not only depend on n but also crucially depend on the coverage parameters. The problem is that different methods depend on different forms of coverage parameters  (e.g., the $\sigma\_{\min}$ term in Theorem 2, $C^\pi$ in Theorem 4, and $C\_\infty^\pi$ in Theorem 5), which are often not comparable. We have a nuanced discussion on this topic in Appendix C.5 for RL theorists readers familiar with the literature.
>
> ---
>
> **In figure 2, can you explain why mb\_naive performs well in the top row but has disastrous error in the last two in bottom row?**
>
> As mentioned on Line 203, this is because mb\_naive strongly biases towards deterministic models when the candidate models have different levels of stochasticity. In the top row of figure 2 all models have similar levels of stochasticity, so this problem is not exposed. In the bottom row, however, columns 2 and 3 correspond to cases where the groundtruth model has relatively high stochasticity, so the bias is fully exposed. In contrast, in column 1, the groundtruth model is the most deterministic one in the family, so mb\_naive’s bias acts in its favor.
>
> Your question (Reviewer Robe also asked the same) made us realize that the bias of mb\_naive is much less known than we thought, so here is a more detailed explanation of the underlying math (which we can add to the appendix): The key is that in the loss we have $\\\| s’ - \tilde{s}\\\|$ (Line 120), where $s’$ is from the data and $\tilde{s}$ from the candidate model. When the candidate is the true model, $s’$ and $\tilde{s}$ are iid, but **their randomnesses are independent and “out of sync”**, which causes the loss to be non-zero and scale with the inherent randomness of the distribution. As a minimal example, suppose the state space is $\mathbb{R}^2$, and we focus on the transition distribution from a particular $(s,a)$ pair where groundtruth is uniform over 4 points $(\pm 1, \pm 1)$. If we calculate the expected loss of groundtruth model itself (on data generated from the same model), we get $1+\sqrt{2}/2$. However, a wrong model that always predicts $(0,0)$ deterministically will get a lower loss of $\sqrt{2}$.
>
>
> ---
>
> **Authors should summarize the theorems briefly in the main text.**
>
> We will surely try to!

---

> > ### Comment · Reviewer_2FCh · 2025-08-01
> >
> > Thank you for your clarifications. I am happy to keep the positive score as it is.

---

### Official Review · Reviewer_G4cF · 2025-07-03

**Clarity:** 2
**Significance:** 2
**Originality:** 2
**Rating:** 3
**Confidence:** 1

**Summary:**

This paper focuses on the model selection for OPE itself. The authors develop: (1) new model-free and model-based selectors with theoretical guarantees, and (2) a new experimental protocol for empirically evaluating them. The authors exemplify the protocol on Gym-Hopper, and find that the proposed new model-free selector, LSTD-Tournament, demonstrates promising empirical performance.

**Questions:**

1. The authors assume the Q function follows a linear function approximation (line 141), which is not w.l.o.g. Are the following proofs based on this assumption?

2. (optional suggestion) OPE and offline RL particularly benefit real-world applications where ground-truth evaluation is not available during the training and OPE process.  The authors would be better to show the effectiveness of the proposed methods in real-world experiments, such as robotics, e-commerce applications.

**Ethical Concerns:**

["NO or VERY MINOR ethics concerns only"]

**Final Justification:**

As I am not familiar with the BVFT, my confidence in accepting or rejecting this paper is low. However, I think the paper writing needs to be improved. For a person who is not familiar with the related area, it is hard to follow.

**Limitations:**

yes

**Quality:**

2

**Strengths And Weaknesses:**

**Strength:**
1. The chicken-and-egg question that the paper focuses on sounds interesting.
2. The paper improves the BVFT method and proposes two detailed methods.

**Weakness:**

1. Assumption Concern. The authors assume the Q function follows a linear function approximation (line 141), which is not w.l.o.g.

2. Unclear Background. The basic idea of BVFT and the double sampling problem should be introduced in more detail and in an easier-to-understand way, e.g., in the Appendix. The readers may have no background knowledge of BVFT.

---

> ### Author Rebuttal · Authors · 2025-07-27
>
> We thank the reviewer for the comments. Unfortunately the reviewer **misunderstood a major technical point** of the paper regarding linear function approximation, which we explain below. We then respond to your other comments.
>
> ---
>
> **We do NOT assume Q functions follow linear function approximation.**
>
> Line 141 is explaining the background knowledge of LSTDQ, which is indeed a method for learning Q-functions with linear function approximation (LFA). However, when we apply it to the model selection problem, **we do not make any assumptions on the Q-functions we select from**. They can be obtained by training neural networks using FQE on offline data (Figure 5 Left), or in our protocol, implicitly defined as the Q-function of the simulator MDP (Figure 5 Right). In any of these cases and in our theoretical development, we do not assume these candidate Q-functions to be generated/learned via LFA. The reason LFA is mentioned is because, when we select among the candidates, we use LSTDQ as a subroutine. LSTDQ indeed requires linear features, and these features are constructed from the candidate Q-functions themselves (Eq.6). Essentially we **reduce** the problem of selecting from **general candidate Q-functions** (which is difficult and very few solutions exist) to the subproblem of **learning with linear features** (which is well studied and has algorithms with strong guarantees), and how to do this reduction (without relying on additional hyperparameters or side knowledge) is the key innovation of our method. Also, if you skip the derivations, treat it as a blackbox, and jump right to the final loss (Eq.(10)), you will see that the equation is simple, closed-form, and has nothing to do with LFA. You asked whether the “proofs [are] based on this assumption” – as you can tell from the paper, no we don’t have this assumption in our main theoretical results (see Theorem 2 for example).
>
> ---
>
> **Background of BVFT and double-sampling**
>
> BVFT is a complicated method and counterintuitive in many ways, and it takes very nontrivial efforts to grasp its main idea even for experts in offline RL theory if they have not seen the work before. So unfortunately we don’t think we can do justice to the method in the limited space. Our paper is written in a way that is meant to be self-contained, and theorists should be able to follow our derivations without needing to know BVFT in the first place, even if we did take the high-level spirit of BVFT (as acknowledged on Line 132) when designing our method. Also, we have provided theoretical and empirical comparisons to BVFT even if the readers don’t know the latter; for example, Line 53 and 173 mention that we have improved rates compared to BVFT, which can be appreciated without knowing the details of the method.
>
> For double sampling, this is a well-documented issue first observed in 1995 [7] and also discussed in Sutton & Barto’s textbook [50] (see Chapter 11.6 in 2nd ed). For a concise explanation, see Section 3.1 of [9]. Again, for the purpose of our paper, the readers just need to know that TD-sq is biased, which we mentioned.
>
> ---
>
> **Experiments in real-life applications**
>
> We agree that this should be what the OPE community strives for. Unfortunately, 99% of the OPE works (in fact we don’t know *any* work that does it) don’t do that for understandable reasons: unlike most of ML where realistic experiments only require realistic data (e.g., ImageNet), RL experiments on real-life applications require engaging with the actual system. For OPE specifically, if we only have off-policy data, how do we know the groundtruth performance of the policy? The only way to get reliable groundtruth is to **run the target policy in the real-system**, and the difficulties in doing so is usually why OPE is needed in the first place. Even if technically possible, this requires closely working with domain experts and obtain access to real-life systems where decisions can cause financial and societal consequences (e.g., in online recommendation, you need to work closely with the company and get their permissions, which often requires an official affiliation with the company), which is a resource not many researchers have access to. All that said, we agree with the sentiment and believe the community as a whole should make efforts to get closer to real-life systems, though this is perhaps a luxurious request at the moment.

---

> ### Comment · Reviewer_G4cF · 2025-08-04
> **Thank for your repsonse**
>
> Thank you for the response.  I am OK with your point 3. In your point 1, you do not assume the linear function approximation, so Theorem 1 in the paper is an analysis of the previous algorithm?
>
> About point 2, I have no further comments as I am not very familiar with BVFT. Regarding the paper writing, I think a clearer statement or organization of the paper would be helpful. Perhaps describing the method you proposed first, and then giving the theorem. (No further changes are required for the paper writing in this discussion, as the reviewer cannot fully know the detailed changes. Take this as a suggestion.)

---

> > ### Author Response · Authors · 2025-08-04
> >
> > Re Theorem 1: you are right (up to some nuances explained below) that Theorem 1 is largely an analysis of an existing algorithm, LSTDQ (Line 139). It essentially serves as a lemma for proving Theorem 2, which is the main theoretical result of the section. That said, we are using a loss-minimization variant of LSTDQ, which is, as far as we know, not considered in the literature, so we have to perform this analysis ourselves. In addition, introducing Theorem 1, which is relatively simple and easy to understand, can help readers understand the more complicated Theorem 2, since the latter is based on the former and shares similar structures.
> >
> > We also thank the reviewer for the additional comments on writing and will consider for revision.

---

> > > ### Comment · Reviewer_G4cF · 2025-08-04
> > >
> > > Thank you for the response. I will maintain my score with low confidence.

---

> > > > ### Author Response · Authors · 2025-08-04
> > > >
> > > > Dear reviewer G4cF,
> > > >
> > > > Thank you for your further response. We would appreciate if you can re-consider your score given that we have clarified one of your 2 listed weaknesses, and the algorithm applies to much more general settings (general Q-functions) than you originally suspected (linear FA).
> > > >
> > > > We understand that your concern about paper structure remains, and want to elaborate further about your suggestion of “describing the method you proposed first, and then giving the theorem”. This is the order in our current presentation, but we guess what you mean (happy to be corrected if we misunderstand your point) is to provide a minimal description of the algorithm (Eq.10 in our case) and its guarantees, before delving into the lengthy explanations of derivations and proof sketches. This is indeed the most standard presentation template for theory papers, and sometimes one can even skip the algorithm description since that is viewed as just the means for enabling the final guarantee. This template has the advantage that, readers without background to understand the derivation details and proof ideas can sometimes still appreciate the significance of the results.
> > > >
> > > > While we would want to embrace this standard presentation style, our results are a bit unusual that makes this difficult. In particular,  our Theorem 2 depends on the minimum singular values of the $A\_{i,j}$ matrices, which are the coverage parameters of the LSTDQ subroutimes in the algorithm. They are deeply intertwined with the derivation process and require background knowledge of LSTDQ to interpret (eg how $A\_{i,j}$ arises from the $\phi\_{i,j}$ directly follows from the quantities defined in the LSTDQ algorithm). This “technical messiness”, which we discuss in Appendix C.5, is an unfortunate consequence of model-free guarantees with only realizability, which uses much weaker assumptions than those cleaner results seen in the literature. Therefore, directly introducing the guarantee (Theorem 2) and the algorithm (Eq.10) without the derivation process unfortunately makes little sense in our context, and we choose to walk the readers through the derivations, essentially structuring the section like a “research log” instead if a top-down theoretical presentation. The style is not uncommon in OPE papers with theoretical components, and a prominent example is the works on the DICE family of estimators (eg DualDICE). We understand that such a style can be hard to follow for readers without the needed background to decipher the derivation process, and still choose it after a careful consideration of the trade-offs.
> > > >
> > > > Nachum et al. NeurIPS-19. DualDICE: Behavior-Agnostic Estimation of
> > > > Discounted Stationary Distribution Corrections.

---

> ### Comment · Reviewer_G4cF · 2025-08-04
>
> Thank you for the further explanation. I can further lower my confidence score. I think perhaps whether the concerns of Reviewer Robe and other reviewers are well addressed matters for the acceptance of this paper.

---

> > ### Author Response · Authors · 2025-08-05
> >
> > Dear Reviewer G4cF,
> >
> > Thank you very much for your openness. We totally understand that concerns of other reviewers (e.g., Reviewer Robe, with whom we are still discussing) can be valid reasons for rejection. That said, for best reviewing practices, we believe every reviewer should form independent opinions. This doesn't mean that we can't use others' criticisms as reasons for rejection, but rather that if we do so, we should read the technical criticisms and make a judgement based on our own understanding.
> >
> > Reviewer Robe's main technical criticisms are based on a less common variant of LSTDQ, which we guess most readers aren't even aware of. (The lack of popularity is not our sole justification against such criticisms; we have more concrete technical arguments too.) If you feel it looks like a legitimate concern but do not understand the technicality yourself, we hope you form your assessment based on aspects you are able to confidently assess. We trust the Area Chair to synthesize all reviewer feedback to form a final decision, and they can always reject the paper if they find Reviewer Robe's technical criticisms critical. Of course, if it's the other aspects of the paper that you are mainly concerned about, we are happy to have further discussions with you to address them.
> >
> > thanks,
> >
> > Authors

---

### Official Review · Reviewer_Robe · 2025-07-03

**Clarity:** 1
**Significance:** 2
**Originality:** 3
**Rating:** 3
**Confidence:** 4

**Summary:**

This paper attempts to address the problem of model selection for off-policy evaluation (OPE) algorithms. It introduces a hyperparameter-free model-free selector named LSTD-Tournament, derived from the LSTDQ algorithm, and provides an accompanying theoretical analysis. The paper also proposes several new model-based selectors and a new experimental protocol for evaluating OPE selectors. This protocol generates candidate Q-functions by perturbing the parameters of a ground-truth simulator. Experiments on the Gym-Hopper environment are presented to evaluate the proposed methods.

**Questions:**

1. Could you clarify the motivation behind the design of the new model-based selectors? What advantages were they hypothesized to have?
2. Regarding the claim that the naive model-based selector fails in stochastic environments, could you elaborate on how Figures 2 and 3 demonstrate this? A more direct explanation in the text would be very helpful, as the visual evidence alone is not entirely clear.
3. Could you comment on the generalizability of your results beyond the Hopper environment and DDPG-based policies?
4. The LSTD-Tournament loss uses candidate functions $Q_k$ or the normalized difference to weigh the Bellman residual of $Q_i$. Could you provide a more concrete intuition for why this is a good design choice? Have you considered or experimented with other weighting functions to verify that using the candidate Q-functions themselves is critical to the method's success?

**Ethical Concerns:**

["NO or VERY MINOR ethics concerns only"]

**Final Justification:**

This paper presents a novel selector, LSTD-Tournament, for a significant problem in offline RL. The discussions with the authors were productive and helped clarify their contributions and my concerns. While the authors provided a clear explanation for the failure of the `mb_naive` baseline, several major issues remain unresolved.

My recommendation remains **Borderline Reject**. The paper contains a promising direction but is undermined by fundamental issues in its claims, theoretical framing, and presentation.

Here is a summary of the unresolved issues that form the basis of my recommendation:

* **Overstated "Hyperparameter-Free" Claim (High Importance):** This is the most critical unresolved issue. The authors' central claim that LSTD-Tournament is "hyperparameter-free" is not accurate. The choice to set the weighting function equal to the feature basis is a significant design decision that functions as an implicit, high-impact hyperparameter. The authors' defense rests on the "on-policy boundedness" property of this choice, but this property's relevance in the challenging off-policy setting is not well-established and does not preclude the potential superiority of other weighting functions. Ignoring this vast design space weakens the paper's core premise.

* **Poor Clarity and Organization (High Importance):** The paper is exceptionally difficult to read. This concern was not adequately addressed and was echoed by other reviewers. The chaotic organization, disruptive forward references, and lack of clear motivation for key components make it a significant challenge for the reader to understand the work and appreciate its contributions.

* **Disconnected Theory and Unmotivated Model-Based Section (Moderate Importance):** The theoretical guarantees for the model-free (Theorem 2) and model-based (Theorems 3 & 4) selectors are built on different, incomparable assumptions. This makes it impossible to theoretically assess if or when the model-based methods should outperform the model-free one. Given their poor empirical performance, this section feels like a collection of unmotivated and inefficient algorithms. The value of reporting these "negative results" is diminished because the analysis does not explain *why* these theoretically sounder methods failed.

In conclusion, while the LSTD-Tournament algorithm is interesting and the paper has valuable technical components, the combination of an overstated central claim, a disconnected theoretical narrative, and severe clarity issues prevent me from recommending acceptance. The paper requires substantial revision to address these foundational problems.

**Limitations:**

Yes.

**Quality:**

2

**Strengths And Weaknesses:**

## Strengths
- **Relevant Problem Formulation**: The paper focuses on a significant and challenging problem in practical offline RL: how to select among different OPE estimators or their configurations. This is a critical step for reliable policy selection before deployment.
- **Solid Theoretical Guarantees**: The LSTD-Tournament selector is supported by a clear theoretical analysis (Theorem 2), demonstrating a standard $n^{−1/2}$ convergence rate, a notable improvement over BVFT.
- **An Attempt at a Better Protocol**: The proposed experimental protocol tries to solve a real problem with existing benchmarks that rely on unstable FQE instances. The idea of generating candidates from perturbed ground-truth models is an interesting direction for creating more controllable testbeds.
## Weaknesses
1. **Clarity and Organization**: One of the paper's primary weakness is its poor organization and clarity, which makes it extremely difficult to follow. The flow of information is chaotic and confusing.
    - *Disruptive Forward References*: The text is littered with forward references that disrupt the reading flow and require the reader to jump to later parts of the paper (or the appendix) before the necessary context is provided. For instance:
        - In the Preliminaries (Section 2), the text directs the reader to "see Figure 5 for a visualization", but Figure 5 is in the appendix and illustrates the experimental protocol from Section 5.
        - Section 3.1 refers to "e.g., \mathcal{G} in Section 4", a concept not yet explained.
        - Section 5.2 (Computational Efficiency) tells the reader to "see Section 6.2" for an example of its application, jumping ahead of the entire experimental setup.
    - *Vague Justification of Claims*: Some claims are not adequately explained. For example, when discussing the naive model-based baseline, the paper claims there is "empirical evidence of this in Section 6 (see `mb_naive` in Figures 2 and 3)" but provides no further interpretation. Looking at the plots, it is not immediately obvious how they serve as damning evidence; in several high-noise environments (e.g., top row of Figure 2), `mb_naive` appears to perform quite well or at least comparably to other methods.
    - *Minor Typos*: There are some typos, such as "textitunits" in Section 5.1.
2. **Misleading "Hyperparameter-Free" Claim and Unmotivated Design**: The paper's central claim of LSTD-Tournament being "hyperparameter-free" is questionable. The design of the feature map $\phi(s,a)$ (e.g., $[Q_i, Q_j]^T$ or the normalized difference version) is a critical design choice with a significant impact on performance, effectively acting as a hyperparameter. In fact, the weighting function $Q_k(s,a)$ in Eq.10 can be any function of $(s,a)$, effectively introducing infinitely many degrees of freedom. The paper acknowledges variants (Section 3.2, Appendix E.2) but heuristically fixes one for the main experiments, which merely hides this degree of freedom rather than eliminating it. The ablation study in Appendix E.2 demonstrates that the performance of LSTD-Tournament is quite sensitive to choice of the weighting function. This core design choice lacks a strong intuitive motivation and is not properly justified through ablation studies comparing it with simpler alternatives, e.g., random functions of $(s,a)$.
2. **Underdeveloped Model-Based Contribution**: The section on new model-based selectors (Section 4) feels extraneous and not well-motivated.
    - *Poor Performance and High Cost*: These selectors are empirically shown to be outperformed by the simpler, more efficient, and more broadly applicable model-free LSTD-Tournament. They also have a higher computational cost ($O(m^2)$), making them less practical.
    - *Lack of Motivation*: The paper offers little intuition for the design of these new selectors (especially the "Sign-flip Selector"). It is unclear what benefits they were intended to provide over existing approaches or the model-free methods.
3. **Limited Diversity in Experiments**: The generalizability of the empirical findings is constrained by a lack of diversity in the experimental setup.
    - *Single Environment*: All experiments are conducted exclusively on the Hopper-v4 environment. Although the authors introduce variations by modifying parameters like gravity and noise, the underlying task structure remain the same. Testing on a wider range of distinct environments would significantly bolster the claims of robustness.
    - *Homogeneous Policies and Datasets*: The target policies are all generated from checkpoints of a single algorithm, DDPG. The behavior policies are also created in a uniform way, typically by adding Gaussian noise to a DDPG policy. A more realistic scenario would involve selecting between policies generated by different types of algorithms (e.g., CQL, IQL, TD3) and using offline datasets collected from varied sources, which this study does not cover.

---

> ### Author Rebuttal · Authors · 2025-07-27
>
> Thank you for your detailed and helpful feedback! We will refer to your listed weaknesses as W1-4 and questions as Q1-4.
>
> ---
>
> **W2: Misleading "Hyperparameter-Free" Claim and Unmotivated Design**
>
> We first address a potential **technical misunderstanding** before responding to your comments.
>
> > **”the weighting function $Q\_k(s,a)$ in Eq.10 can be any function of $(s,a)$, effectively introducing infinitely many degrees of freedom.”**
>
> This is **incorrect**. In particular, choosing any function, such as a purely random function of $(s,a)$ (more on this below), could lose the guarantees of Theorem 2. As we mentioned on L180, only “non-degenerate **linear transformations**” of $[Q\_i, Q\_j]$ enjoy the same kind of guarantees. **Choosing an arbitrary weighting function unrelated to $Q\_k$ simply cannot be called a variant of LSTD-tournament.**
>
> We hope this clarification eases your concerns. Below we respond to your other related comments, assuming you still stand by them after this clarification. As you have noted, we acknowledge this degree of freedom (**the linear transformation**) and provide a recommended choice (normalized diff). That said, there is a big difference between this kind of “small” hyperparameters vs. the “big” ones we mention in the introduction (e.g., neural architecture of FQE; L35). To put it simply,
>
> > Are we comfortable recommending neural nets of 2 hidden RELU layers with 100 nodes per layer whenever someone applies FQE in _any OPE problem_? **Of course not!**
>
> > Are we comfortable recommending the normalized diff version whenever someone applies LSTD-tournament to _any OPE model-selection problem_? **Yes! (see reasons below)**
>
> **lacks motivation for the design, and ablation in App E.2 demonstrates [sensitivity]**
>
> There is good motivation for normalization. In particular, imagine if a wrong candidate $Q\_k$ has enormous magnitude. When such a $Q\_k$ serves as the weighting function, it can artificially blow up the loss of other $Q\_i$, including the true Q-function, and the proof of our theoretical guarantee relies on true Q having a reasonably small loss. Therefore, controlling the magnitude of weighting functions via normalization is well-motivated. Taking difference is, indeed, more heuristic, but its use in RL to help with numerical stability is also well-documented [25,10].
>
> The 3 variants do have different performance profiles in App E.2. However, compared to other baselines, the qualitative nature of their performances are not that different, and none of them strictly dominate the others. Given that their performance comparison is largely a tie, and there is good reason to prefer normalization (to protect against large Q) and difference (for numerical stability), we feel comfortable recommending the normalized diff version as a **universal choice**. Fixing such a choice, in our opinion, is _exactly eliminating the degree of freedom_ rather than “hiding it”. For example, BVFT also has a discretization error as a hyperparameter, and the authors provide a heuristic to pick it in [61], which we adopt in our implementation.
>
> ---
>
>
> **Q4: why are $Q\_k$ good weighting functions intuitively?**
>
> Using $Q\_k$ as weighting functions purely emerges from our theory. We don’t have more intuitive explanations other than the theory, which we don’t view as a negative; in fact, BVFT is just as counterintuitive as (if not much more than) LSTD-tournament.
>
> Popping up a level, most RL research follows an “empirical-first” methodology of “**design intuitive algorithms -> test empirically -> if works, find theoretical justifications**”, which is perhaps why the reviewer asks this question. In contrast, we start from derivations with strong theoretical guarantees (Theorem 2) as the basis of research. This methodology has the unique advantage of tapping into those “counterintuitive” ideas in the algorithmic space. If not for it, algorithms like BVFT will perhaps never be found.
>
> **W2: How about other weighting functions, such as “random functions of (s,a)”?**
>
> Perhaps surprisingly, if you use “random functions” as weights, the loss becomes degenerate. In particular, we can write Eq.10 as
>
> $$ \max_{w \in  \mathcal{W}} |\mathbb{E}\_{\mu}[w(s,a) (Q\_i-\mathcal{T}^\pi Q\_i)]|,$$
>
> As you pointed out, LSTD-tournament uses $Q\_k$ as the weighting function $w$. The most random way to generate $w$ is to generate an i.i.d. weight scalar for each $(s,a)$ in the dataset. Suppose this i.i.d. distribution has mean $C$. Now, because the randomness in weight is independent of everything else, the outside expectation will effectively first marginalize out such randomness, which means the expected loss is simply equal to
>
> $$|\mathbb{E}\_{\mu}[C (Q\_i-\mathcal{T}^\pi Q\_i)]|.$$
>
> When $C \ne 0$, this coincides with “average Bellman error” (L290) which we include in the experiments. It has poor theoretical properties: even if a candidate $Q$ has the wrong shape, you can always add a constant to it–which does not change its shape–to make its average Bellman error $0$.
>
> If we want make the weights “nontrivially random”, we need to leverage information in $(s,a)$, e.g., generate a random function that is smooth over $S \times A$ w.r.t. some metric. However, the choice of the metric and how exactly the function should be generated are “big hyperparameters” that are left as design choices to the user.
>
> Moreover, the sign-flip selector (App B.2) also fits this form, where $sgn(Q\_i - g)$ serves as weighting functions. So that can also be viewed as a comparison where the weighting functions are chosen differently.
>
> ---
>
> **Q2 \& W1: explain the failure of naive\_mb**
>
> As mentioned on L203, naive\_mb has a strong bias towards more deterministic systems. That is, **when the candidate models have varying degrees of stochasticity**, naive\_mb tends to choose the least stochastic one. You observe that naive\_mb performs well in the top row of Fig 2, and that’s because all candidate models in MF.G have the same degree of stochasticity. (There is a potential confusion: “MF.G” and “MF.N” are specified on L271; the value of $g$ and $\sigma$ after “:” describes $M^\star$, **not** “MF.G” or “MF.N”.) **It’s the relative difference in stochasticity among candidate models that matter, not the absolute level.** This also explains why naive\_mb performs particularly well in the left plot of bottom role, since the groundtruth model has the least stochasticity and naive\_mb’s bias happens to play in favor.
>
> We believe this particular bias is well known to the theory community (since the underlying issue is a strong motivation for widely used tools such as Wasserstein distance) and thus omitted a detailed explanation. Given your feedback, however, we will add more discussion. The key is that for $\\\| s’ - \tilde{s}\\\|$ in the loss (L120), $s’$ is from the data and $\tilde{s}$ from the candidate model. When the candidate is the true model, $s’$ and $\tilde{s}$ are iid, but **their randomnesses are independent and “out of sync”**, which causes the loss to be non-zero and scale with the inherent randomness of the distributions. As a minimal example, suppose the state space is $\mathbb{R}^2$, and we focus on the transition distribution from a particular $(s,a)$ pair where groundtruth is uniform over 4 points $(\pm 1, \pm 1)$. Loss of groundtruth model itself is $1+\sqrt{2}/2$, which is higher than a wrong model that always predicts $(0,0)$ deterministically ($\sqrt{2})$.
>
> ---
>
>
> **Q1 \& W3: Motivation for model-based selectors**
>
> We understand the sentiment that the model-based selectors feel “extraneous”. In fact, we received similar comments in a previous round of review and decided to defer the contents to the appendix, only leaving a short summary in the main text.
>
> That said, we argue that there is strong **theoretical** motivations for the model-based selectors. In particular, the selectors can leverage the information about Bellman operators (L207), which is accessible in the model-free setting, and obtain standard and elegant theoretical guarantees (Theorems 3 and 4) familiar to the RL theory literature. Our intuition is that **methods that properly leverage additional side information often performs better**, and the “standardness” and “elegance” of the guarantees are, in our opinion, enough motivation to consider the model-based selectors.
>
> In fact, before we implement the model-based selectors, we were **fully anticipating** a trade-off between performance (that model-based should _outperform_ model-free) and applicability/computational cost (model-free applies to more settings and is cheaper to run). **The results come out as a big surprise, and we think reporting such negative results is invaluable to the research community.** This gets back to our comment on the research methodology: you said that the model-based methods lack motivation _because_ it performs poorly, which reflects the “empirical-first” methodology. In our theory-first approach, we develop the methods in theory and decide to include it **before we do any experiments**: if the experiments confirm our hypothesis, great; if not (as we saw), the surprisingly poor performance of model-based selectors further highlights the effectiveness of LSTD-tournament. As for why model-based selectors fail, we have some speculations (see response to ZupL’s (3)).
>
> ---
>
> **Q3: Generalizability beyond Hopper and DDPG policies**
>
> This is a valid point, and as usual, the more experiments on more environments/settings, the always better. That said, unlike most of RL research where training algorithms have many small design choices that can be tweaked to overfit to a setting, our loss is closed-form and optimization-free, leaving very little room for overfitting.
>
> ---
>
> **Q1: Sec 2 pointer to Fig 5 describes Sec 5 protocol**
>
> Fig 5 left describes the practical setup, which directly corresponds to the text in Sec 2.

---

> > ### Comment · Reviewer_Robe · 2025-08-04
> > **Comment on the Rebuttal I**
> >
> > Thank you for your detailed response and for engaging with my feedback. While the rebuttal clarifies some aspects of your work, it also reinforces and, in some cases, deepens my original concerns, particularly regarding the theoretical justification for key design choices and the overall clarity of the paper.
> >
> > I will address the most critical points below.
> >
> > **1. On the Degrees of Freedom in the Weighting Function (W2/Q4)**
> >
> > I must first address what you term a "technical misunderstanding." Your rebuttal states it is "incorrect" that the weighting function can be any function of $(s,a)$ and that only "non-degenerate linear transformations" of $[Q_i, Q_j]$ would preserve the guarantees.
> >
> > This is, with all due respect, a mischaracterization of the underlying theory. The LSTD framework is more general than presented. Let's define a feature function $\phi(s,a)$ (the basis, e.g., $[Q_i, Q_j]^T$) and a separate test function $\varphi(s,a)$ (the weighting function). The generalized LSTD equation is $A_{\varphi\phi}\theta = b_{\varphi}$, where:
> > * $A_{\varphi\phi} = \mathbb{E}[\varphi(s,a) (\phi(s,a) - \gamma \phi(s',\pi))^T]$
> > * $b_{\varphi} = \mathbb{E}[\varphi(s,a) r]$
> >
> > For the true parameter $\theta^\star$ satisfying $Q^{\pi}$  = $\phi^{T}\theta^\star$, the Bellman equation implies $\phi(s,a)^T\theta^\star$ = $\mathbb{E}_{r,s'|s,a}[r + \gamma \phi(s',\pi)^T \theta^\star]$. Substituting this into the definition of $A_{\varphi\phi}\theta^\star$ shows that $A_{\varphi\phi}\theta^\star = \mathbb{E}[\varphi(s,a)r] = b_{\varphi}$. This fundamental equation holds for an **arbitrary** choice of test function $\varphi(s,a)$.
> >
> > Therefore, your choice to set the test function equal to the feature function ($\varphi = \phi = [Q_k, Q_i]^T$) is a **specific design choice**, not a theoretical necessity. This choice allows you to directly leverage the existing proof of Theorem 1 for the standard LSTDQ algorithm. However, it does not eliminate the vast space of other valid choices for $\varphi$; it merely selects one specific point within that space. My original critique stands: this introduces a significant degree of freedom that is not fully justified, and your claim that my understanding is "incorrect" seems to stem from a conflation of your specific implementation with the general theoretical principle.
> >
> > Your dismissal of "random functions" by analyzing a trivial i.i.d. case also does not address the core of my point. My suggestion was not about a specific random construction but about interrogating the necessity of using $\{Q_k\}$ versus any other function from the vast space of valid test functions. The fact that the sign-flip selector uses a different weighting function ($sgn(Q_i - g)$) further proves that this design space exists, yet a comparative theoretical analysis to show why one choice might be superior to another is absent.
> >
> > **2. On Motivation, Intuition, and the "Theory-First" Methodology (W2/Q4, Q1/W3)**
> >
> > I must clarify a misinterpretation of my review's perspective. My request for "intuition" and "motivation" is not a preference for an "empirical-first" methodology, nor is it a disregard for theory. On the contrary, I believe that strong theory should provide deep **insight**. It should not just yield a formula, but also explain *why* a particular design works and what makes it powerful compared to alternatives.
> >
> > My critique, for both the model-free and model-based selectors, is that this insight is lacking. The argument that "Using $Q_k$ as weighting functions purely emerges from our theory" is not a sufficient justification. The theory shows it is a *valid* choice, but not necessarily the *only* or *best* choice. Why is this specific choice superior to other valid alternatives within the same theoretical framework? What properties of the problem does it uniquely leverage? This is the motivation I found missing.
> >
> > Similarly, I did not state they "lack motivation *because* it performs poorly." I stated they "offer little intuition." Reporting negative results is indeed valuable, but their value is maximized when we understand the hypothesis that was being tested and why it might have failed.

---

> > ### Comment · Reviewer_Robe · 2025-08-04
> > **Comment on the Rebuttal II**
> >
> > **3. On Clarity and Other Concerns (W1, Q2, Q3)**
> >
> > I appreciate the detailed explanation regarding the naive model-based selector's bias towards deterministic systems. This explanation is clear, plausible, and helpful. It is also precisely the kind of detailed interpretation that was missing from the original manuscript, which highlights my original point about the paper's poor clarity. My comment on the disruptive forward references and chaotic organization, which you did not address, remains a primary concern.
> >
> > I also concede your point that a closed-form, optimization-free loss function is less susceptible to overfitting in the traditional sense. However, this does not fully address the concern about generalizability. The dynamics, reward structures, and policy complexities can vary dramatically across different RL environments, and demonstrating robustness on a single environment family (Hopper), even with parameter variations, is still a limitation.
> >
> > In summary, while I thank you for the engagement, my primary concerns remain. The rebuttal has revealed a deeper issue regarding the justification of the core theoretical claims. The argument that the specific choice of weighting function is a necessity rather than a design choice is not technically sound, and the defense of this choice lacks a compelling theoretical or intuitive foundation beyond "it allows us to use Theorem 1." Combined with the unaddressed issues of clarity and limited experimental diversity, I must stand by my original assessment.

---

> > > ### Author Response · Authors · 2025-08-05
> > >
> > > Dear reviewer Robe,
> > >
> > > As the time for author-reviewer discussion runs out, we'd very much appreciate your further consideration and comments on our replies to your comments to allow for sufficient exchange of ideas and opportunities for us to address your concerns.
> > >
> > > As a quick recap, your criticisms focus on theoretical justifications, but you have commented very little on the core theoretical results, namely the guarantees (Theorem 2 for LSTD-tournament, and Theorems 3 and 4 for model-based selectors). We want to emphasize that obtaining such guarantees are **highly-nontrivial** (especially for the model-free setting) and are the theoretical basis of our work. You mentioned multiple times that we can just "use other weight functions", and seem to indicate that they are just as good as standard LSTDQ in every aspect. You did not, however, respond to our counterargument, that using general $\psi$ may not enjoy the kind of guarantee in Theorem 2. It's true that we can still obtain a result superficially similar to Theorem 2 for the $\psi$ variant, but the matrix singular value will not be guaranteed to be bounded in the on-policy case, which violates the minimal condition for being a coverage parameter in OPE, let alone the "infinitely many degrees of freedom" it introduces (you treated it as a criticism, but we think this just further highlights standard LSTDQ as a reasonable choice).
> > >
> > > We also showed above that the normalized diff variant is not derived from $\psi$-version of LSTDQ, but rather standard LSTDQ with different choices of $\phi$. This misunderstanding perhaps led you to think that "the authors already tapped into the vast space of possibilities and there is no reason they don't look further", which is technically not the case.
> > >
> > > thanks,
> > >
> > > Authors

---

> > > > ### Comment · Reviewer_Robe · 2025-08-06
> > > >
> > > > Dear Authors,
> > > >
> > > > Thank you for the engaging and detailed exchange throughout this discussion period. I appreciate you taking the time to recap your position. Having considered your arguments, I would like to offer my concluding assessment.
> > > >
> > > > I recognize the technical work involved in deriving the guarantees for your proposed algorithms, particularly Theorem 2 for LSTD-Tournament. My remaining concerns, however, have been clarified through our discussion and center on several key aspects of the paper that I believe require substantial revision.
> > > >
> > > > **1. On the "Hyperparameter-Free" Claim**
> > > >
> > > > My primary concern is the overstatement of the "hyperparameter-free" claim. While I understand that setting the weighting function equal to the feature basis ($\psi=\phi$) is the standard approach for LSTDQ and ensures the desirable on-policy boundedness property you highlighted, the relevance of this property in the challenging *off-policy* setting is not as clear-cut.
> > > >
> > > > As has been explored in some corners of the literature (e.g., in minimax approaches to OPE [1]), the choice of weighting function is a critical design lever. In off-policy scenarios with poor coverage, the standard choice of $\psi=\phi$ may not be robust, and using more flexible or even adversarially chosen weighting functions could be necessary to obtain reliable estimates. Therefore, the decision to use the standard LSTDQ formulation is not a given; it is a significant design choice with a potentially large impact on performance. This choice effectively constitutes a crucial, implicit **hyperparameter**. Acknowledging this would more accurately frame the contribution of LSTD-Tournament.
> > > >
> > > > **2. On the Disconnected Theory and Model-Based Selectors**
> > > >
> > > > My second point relates to the theoretical structure of the paper. While I acknowledge the non-trivial nature of the proofs for Theorems 3 and 4, these results feel disconnected from the model-free analysis. The guarantees for the model-based selectors (relying on standard concentrability) and the model-free selector (relying on $\sigma_{min}(A)$) are based on different and incomparable assumptions.
> > > >
> > > > This creates two issues. First, the paper lacks a unified theoretical narrative that would allow us to understand if or when the model-based approach offers a *provable* advantage over the model-free one. Without this, the motivation for introducing these specific model-based selectors is weak. Second, their empirical failure, combined with the disconnected theory, leaves the reader with little insight. The value of reporting a negative result lies in understanding *why* the approach failed. The current analysis does not provide this. Without a clear theoretical or empirical justification for their potential superiority, these selectors appear to be statistically and computationally inefficient designs.
> > > >
> > > > **3. On Clarity**
> > > >
> > > > Finally, I must reiterate that the paper's **clarity and organization** remain a significant barrier. This was a primary point in my initial review, and I note that other reviewers have expressed similar concerns. The flow of information and the disruptive forward references make the paper exceedingly difficult to parse, obscuring the paper's core contributions.
> > > >
> > > > **Summary and Path to Acceptance**
> > > >
> > > > In summary, while the paper contains a promising algorithm in LSTD-Tournament and non-trivial technical components, I believe it requires significant revision before it is ready for acceptance. The key areas for improvement are:
> > > >
> > > > 1.  **Revisiting the "hyperparameter-free" claim (High Importance):** The paper should abandon this overstatement and instead frame the choice of weighting function as a motivated design decision, discussing its implications.
> > > > 2.  **Improving clarity and organization (High Importance):** A thorough restructuring is needed to present the arguments and results in a clear, linear, and accessible manner.
> > > > 3.  **Strengthening the model-based section (Moderate Importance):** The paper would be much stronger if it either developed a more integrated theory that allows for direct comparison or provided a deeper analysis of why these methods failed, offering a clear scientific takeaway for the community.
> > > > 4.  **Broadening experimental validation (Lower Importance):** While secondary to the points above, testing on a more diverse set of environments would bolster the empirical claims of robustness.
> > > >
> > > > Thank you again for the thorough and professional discussion. I wish you the best with your work.
> > > >
> > > > Sincerely,
> > > > Reviewer Robe
> > > >
> > > > **References**
> > > >
> > > > [1] Uehara, Masatoshi, Jiawei Huang, and Nan Jiang. "Minimax weight and q-function learning for off-policy evaluation." International Conference on Machine Learning. PMLR, 2020.

---

> ### Author Response · Authors · 2025-08-04
>
> We thank the reviewer for your detailed explanation. We now understand your concern about "any weighting function $w(s,a)$ suffices". Indeed we are aware of what you call the "generalized LSTDQ" equation, where the "test function", or sometimes referred to as the "instrument" (which stems from an instrumental-variable (IV) view of LSTD), can be chosen to be other functions. It seems that both of your points 1 and 2 are rooted on this point.
>
> **A. Not a well-studied method.** This is a valid technical point, but as far as we know, this **generalized LSTDQ** is a niche idea that floats around anecdotally among a small part of the community. Based on our understanding of literature, this method is never sufficiently investigated in the LSTD literature. The only reason we are aware of this is due to some conversation with researchers in the community, where "LSTD can be generalized in this way" comes up in casual conversations as a side comment. We won't give names here but our impression is that these authors may have mentioned this generalization lightly in perhaps a handful papers as a pass-by comment. We are pretty surprised that the reviewer brings this variant up and uses it as the foundation for most of your technical criticisms, but perhaps we are wrong about its role in the literature; if this variant has been systematically studied by the RL theory community, we would appreciate pointers to the thread of works on this matter.
>
> **B. Additional hyperparameters and the lack of basic properties.** As we mentioned in the rebuttal, using arbitrary weighting functions may generally lose the guarantee of Theorem 2 (e.g., iid weights). When they don't, there is also the question of additional hyperparameters, e.g., how to choose $\psi$? Note that $\psi=\phi$ is not an arbitrary choice. Not only is this the most dominant choice in the literature, but it also enjoys an "on-policy boundedness" property. That is, we know that $\sigma_{\min}(A)$ serves as the coverage parameter of LSTDQ, and the quantity is lower-bounded away from $0$ when the data is sampled from an invariant distribution of the target policy. Despite the many mysteries around coverage in LSTDQ (discussed in Appendix C.5), this is a basic property that OPE estimators are expected to have (see [21]) and standard LSTDQ enjoys, whereas choosing arbitrary $\psi$ may not.
>
> **C. Our research is self-contained without examining other choices of $\psi$.** We agree with the reviewer that there might be other choices of $\psi$ that make things work better, but our work is still self-contained, justified, and novel without considering them. In particular, another choice of $\psi$ will be worth considering theoretically if it satisfies two conditions
>
> 1. It can be constructed without requiring further information. In our case, the only thing we can rely on is $\\\{Q\_i\\\}$ in the model-free case. (You mentioned the sign-flip estimator as another choice, but that uses the Bellman operator through the $\mathcal{G}$ class and is not applicable to the model-free setting.)
>
> 2. It has theoretical properties better than (or at least as good as) $\psi=\phi$, including the "on-policy boundedness" property.
>
> **We simply aren't aware of any choice of $\psi$ that satisfies both.** Of course exploring alternative choice opens an interesting venue of future research, which we are happy to discuss at the end of the paper. But the **potential existence** of them does not affect our work: our LSTD-tournament is a framework where any improvement in the "base algorithm" (i.e., LSTDQ) can be directly plugged in to produce a better model-selection algorithm. **When theoretically more motivated choices are not obvious---which we believe is the case now---it is only natural to start investigation with the most standard choice widely considered in the literature.**
>
> The reviewer also asks us to "eliminate the vast space" to make it "a theoretical necessity". **We don't understand how this is doable or even necessary for a publication.** As we mentioned above, we believe that better choices can possibly exist, just that we do not know for now. When you design an algorithm, can you only publish it once you prove that no better variants exist? If you were reviewer for all the LSTD papers in the literature (to which your criticisms also apply to), would you reject most of them because they don't study the $\psi$ variant? Perhaps your did not know the "on-policy boundedness" property, but even without it, we feel that the mere fact that $\psi=\phi$ is the standard choice in the literature & no obvious better choice exists should suffice.

---

> ### Author Response · Authors · 2025-08-04
>
> 1. **Model-based explanation:** We are glad you find the explanation clear. As we mentioned, this is a widely known issue (for which the double-sampling problem can be viewed as a manifestation) and one of the main motivations for studying distribution distances such as Wasserstein's. We will include it in the appendix for readers unfamiliar with the issue. If you have other concrete concerns, we are happy to address in a similar manner.
>
> 2. **Forward pointer and organization:** We ran out of space in rebuttal and unfortunately could not respond to these non-technical issues in detail, but we did respond to the forward pointer issue of Figure 5. Note that the left panel of Figure 5 illustrates a practical setting, which directly corresponds to the text in Section 2. If given more space, we plan to bring Figure 5 to the main text, likely where the Section pointer 2 is. We don't think the existence of the right panel (protocol in Section 5) is a big issue, since the two panels are best viewed side by side as a comparison.
>
> **foward pointer to $\mathcal{G}$ in Section 3.1**: this is in the parenthesis and we believe the sentence outside is self-contained ("Common approaches to debiasing this objective involve additional “helper” classes"). The parenthetical comment serves as supporting evidence which readers can skip if they believe the leading sentence, and we could very well cite a paper (e.g., Antos et al. [6]). Since we already discussed this later in the paper ourselves, we thought that a pointer to the later contents saves the readers the effort of looking up the literature.
>
> **Section 5.2 (Computational Efficiency) [refers to] Section 6.2**: again this is a parenthetical comment, which readers can skip without losing the main picture. The computational efficiency is a side comment anyway since most of our paper (both theoretically and empirically) consider the statistical aspects of the algorithms, so a brief discussion of the computational aspects at the end of the protocol section, before the empirical results, seems a natural choice to us.
>
> 3. **Single-environment is still a limitation despite less overfitting**: We completely agree and are happy to include it in the discussion of limitations in the main paper. This is also why Sec 6 is called "Exemplification of the protocol" (not "Experiments") and the introduction also highlights the "preliminary" (L69) nature of our empirical results. Given that the theoretical developments and the design of protocol are also significant contributions, we feel that a preliminary experiment section would be an reasonable supplement.

---

> ### Author Response · Authors · 2025-08-04
>
> **normalized diff is NOT using generalized LSTDQ**: Upon responding to your comments, we realize a subtle technical point. Your main technical concern is about the vast space of $\psi\ne\phi$, and you mentioned that variants like normalized diff belong to it. That is not correct; for all variants considered in our paper, we are always using the same standard LSTDQ algorithm, only with different choices of $\phi$ (up to linear transformations). $\psi=\phi$ always holds. (In fact, if you look at Appendix E.2, we only specify $\phi$ and never need to introduce $\psi$.) It is indeed true that the final loss will look like what you are imagining: that the TD error part remains the same and only the "test function" changes. This is because when we change the definition of $\phi$ by linear transformation, the candidate $\theta$ will also change accordingly (they are no longer [0, 1], [1, 0] in these variants), and in the loss derivations in Eq.7, the $\phi^\top$ and $\theta$ will always recombine into the familiar TD error.

---

> > ### Author Response · Authors · 2025-08-04
> >
> > Apologies if this causes spam --- we realize that you were somehow excluded as a reader in our replies above (3 posts, 2 responding to your "Comment on Rebuttal I/II", and one additional technical follow-up). We just changed the visibility of those posts and wanted to make sure that you are notified of our comments.
> >
> > thanks,
> > Authors

---

> ### Author Response · Authors · 2025-08-06
>
> We thank the reviewer for the consideration of our response and further elaborations. We understand the limited time and efforts the reviewer has on our paper and the intention to make this discussion "final", and we would still like to provide response to your comments for you and other reviewers.
>
> We believe the reviewer's main criticisms (related to revision suggestions **1** and **3**) are due to your unfamiliarity of the RL theory literature.
>
> ---
>
> **the relevance of this [on-policy boundedness] property in the challenging off-policy setting is not as clear**
>
> First, we believe we have successfully convinced the reviewer that,
>
> > **if this property is relevant, it should largely ease your concern on the hyperparameter-free claim**, as it provides a unique theoretical advantage for standard LSTDQ compared to the infinitely many variants the reviewer is suggesting.
>
> **We will greatly appreciate it if you can acknowledge this**, which will make it easier to explain the core issue to other reviewers and the AC and allow them to make independent judgements.
>
> There is **wide consensus** of the significance of the property in the off-policy setting in the offline RL theory community. Basically, while it's insufficient (which is aligned with your description of the off-policy setting as "challenging"), it serves as an important **sanity-check and necessary condition**: in offline RL, the coverage parameter characterizes how the data distribution covers that induced by the target policy. In general, we expect the parameter to be small when the two distributions are close (i.e., on-policy), and gradually increase when the data distribution deviates from the target distribution. $C^\pi$ and $C\_\infty^\pi$ in the model-based guarantees are standard coverage parameters (they also appear in the guarantees of FQI, MWL/MQL, etc) that satisfy this property. **If a guarantee comes with a coverage parameter that doesn't have this property, it will be considered highly problematic and of little relevance.** So our developments of the algorithms follow the rationale of **"obtaining this kind of guarantees is very nontrivial, and we provide 3 ways of doing so"**.
>
> ---
>
> **Disconnected Theory and Model-Based Selectors; more integrated theory**
>
> We agree with the technical point that the coverage parameters between model-free and model-based selectors are incomparable. In fact we explicitly acknowledged this and offered a nuance discussion in Appendix C.5. That said, **the on-policy boundedness property _is_ a framework that can unify both**, though admittedly it is a very loose characterization. While it would be certainly nice to provide a tighter unification, **this is a long-standing open problem in offline RL theory, and it is very unreasonable to expect us to solve it in a paper where this is not the focus.** In fact, this problem is not specific to our work: if you compare the guarantees of LSTDQ in the literature (which is analogous to our Theorem 1) with those of other algorithms (e.g., FQI, MWL/MQL, which are analogous to our Theorems 3 and 4), **you'd reach the same conclusion that they lack a unified framework and are incomparable**. This open problem **is explicitly acknowledged** through a comment on the unusual nature of coverage parameters in LSTDQ and abstractions (compared to those of FQI, MWL/MQL, etc), in a survey that summarizes the recent advances in offline RL theory [21]:
>
> > _This is a general theme with algorithms that only require realizability [such as LSTDQ and state abstractions], that error propagation and coverage are conceptually much more complicated and hard to interpret._
>
> [21] Offline reinforcement learning in large state spaces: Algorithms and guarantees. Jiang and Xie. 2024.
>
> ---
>
> **[weight functions] in minimax approaches to OPE [1]**
>
> Indeed, LSTD-tournament looks very similar to MQL in [1] you cited. We are **very familiar** with the work and intended to comment on it (and did not due to space limit). In MQL, the weight function needs to capture the importance weight function, which requires an additional realizability assumption and thus hyperparameter choices. **This is exactly why algorithms like MQL cannot be considered hyperparameter-free for model-selection** and we have to base our theory on a different analysis. We also acknowledged that better choice of $\psi$ could exist, which is an interesting future direction and shouldn't be treated as a major flaw of our current work.
>
> ---
>
> To conclude, we appreciate your thoughtful comments and believe we fully understand your technical points, and agree that there is room for improving clarity and structure of the paper. However, we stand firmly by our position and **reject the reviewer's criticisms on the "hyperparameter-free" issue** and **the lack of motivation for model-based selectors** (which already have minimized presence in the main text), which form the basis of the reviewer's technical concerns.

---

### Note · Authors · 2025-08-11

Dear review team,

To facilitate reviewer-AC discussion, here is a summary of the discussion and outstanding concerns, mostly from Reviewer **Robe**. While we disagree with their main criticisms, **we will discuss them in the paper for readers' awareness and own judgement.**

---

**Generalized LSTDQ**

Reviewer **Robe**'s main technical criticism is that, one can replace LSTDQ in our algorithm with the "generalized LSTDQ" variant, revealing "infinite degrees of freedoms" that we need to "eliminate to make [standard LSTDQ] a theoretical necessity".

The productive discussion has revealed where our disagreements lie:

0. The "generalized LSTDQ" is largely a **folklore method** lacking systematic investigation. The reviewer provided no canonical reference. **Google scholar search of exact match of "generalized LSTD(Q)" yields no relevant results.**

1. Choosing standard LSTDQ yields a guarantee where the coverage parameter is **well-bounded in the on-policy case**.

2. The variants suggested by the reviewer **lack this property**.

3. The property makes the choice of LSTDQ natural. **While the reviewer disagrees with its relevance,** this property is clearly understood as a necessary condition in OPE theory: A "coverage" parameter must be small when data distribution is close to the target distribution, a.k.a., near on-policy. **Not passing this sanity check makes any guarantee questionable and potentially vacuous, which will be the case for the variants suggested by the reviewer.**

4. We agree that considering these variants makes interesting future directions, but **disagree that we _must_ exhaustively eliminate them in current work**. Research is about making progress, not chasing perfection.

---

**Incomparability between model-free and model-based methods**

We explicitly acknowledged and discussed this in submission (App C.5), which is a **long-standing open problem in theory** mentioned by recent surveys. It is unreasonable to expect us to solve it in the current paper.

---

**Presentation and Clarity**

Multiple reviewers have writing concerns, which we acknowledge. We believe this is partially due to the unusual topic where most readers are not familiar with the complications in the setting, which we tried to clarify with efforts like Figure 5 (appreciated by reviewer **ZupL**). Reviewer **Robe** criticized forward pointers, but those are mostly parenthetical comments which can be skipped on first reading.

---

### Decision · Program_Chairs · 2025-09-17

**Decision:**

Accept (poster)

**Comment:**

This paper addresses the problem of hyper-parameter selection in off-policy evaluation (OPE) which is of critical performance for the safe deployment of RL agents and an important problem for the RL community. Many OPE methods have hyper-parameters that cannot be practically tuned; this motivates the need for hyper-parameter free methods that can identify which of a set of OPE estimates is going to produce the most accurate estimate. This paper is primarily theoretical in nature. It develops two methods: one model-free and one model-based for OPE estimator selection that show improved convergence rates. The paper also claims a new experimental protocal for evaluating the quality of learned value functions and dynamics models and demonstrates this protocal on the Hopper gym environment. The empirical evaluation shows that the new method provides more reliable value function assessments than the prior BVFT method that could be considered the prior leading method for this problem.

The paper makes an important contribution to the theoretical foundations of OPE and addresses a problem of great practical value. Theoretical results go beyond prior work and are correct to the best of the reviewers knowledge. The work seems to overcome limitations of the prior BVFT method and has improved theoretical guarantees.

A couple of concerns were discussed by the reviewers concerning the claims made and presentation of the paper. These are (1) disagreement over the extent to which the method is truly hyper-parameter free and (2) the model-based section of the paper seemed a little disconnected from the rest of the paper. I will add to these concerns that the paper claims a new experimental protocal but it seems very similar to the work "High-confidence error estimates for learned value functions" by Sajed et al. (2018).

Overall, the paper makes a solid theoretical contribution and shows improved empirical performance on an important problem and the reasons to accept are stronger than the limitations raised. All concerns could be addressed in the camera-ready version.

Suggestions to authors:
1. Discuss alternative design decisions regarding the feature map even if these are left to future work.
2. Add more motivation for the model-based estimators.
3. Either tone down the claim about a new experimental protocal or discuss how it differs from prior work. I don't see this as a major contribution of the paper but of course if there are novel elements then please highlight them.